# A perivascular niche for multipotent progenitors in the fetal testis

Deepti L. Kumar[1] & Tony DeFalco[1,2]

Androgens responsible for male sexual differentiation in utero are produced by Leydig cells in the fetal testicular interstitium. Leydig cells rarely proliferate and, hence, rely on constant differentiation of interstitial progenitors to increase their number during fetal development. The cellular origins of fetal Leydig progenitors and how they are maintained remain largely unknown. Here we show that Notch-active, Nestin-positive perivascular cells in the fetal testis are a multipotent progenitor population, giving rise to Leydig cells, pericytes, and smooth muscle cells. When vasculature is disrupted, perivascular progenitor cells fail to be maintained and excessive Leydig cell differentiation occurs, demonstrating that blood vessels are a critical component of the niche that maintains interstitial progenitor cells. Additionally, our data strongly supports a model in which fetal Leydig cell differentiation occurs by at least two different means, with each having unique progenitor origins and distinct requirements for Notch signaling to maintain the progenitor population.

[1] Division of Reproductive Sciences, Cincinnati Children's Hospital Medical Center, 3333 Burnet Avenue, MLC 7045, Cincinnati, OH 45229, USA.
[2] Department of Pediatrics, University of Cincinnati College of Medicine, 3230 Eden Avenue, Suite E-870, Cincinnati, OH 45267, USA. Correspondence and requests for materials should be addressed to T.D. (email: tony.defalco@cchmc.org)

Leydig cells (LCs) are steroidogenic cells present in the interstitial compartment of the testis. They are responsible for the synthesis of androgens required for initial virilization and patterning of the male external genitalia during fetal life and proper male-specific development and spermatogenesis throughout postnatal and adult life. Low testosterone levels in humans have been associated with male reproductive health disorders, such as impaired spermatogenesis, low sperm count, ambiguous genitalia, and male infertility[1–3].

During development in mice, LC specification begins shortly after sex determination at embryonic day (E) 12.5[4]. Thereafter, in rodents the fetal Leydig cell (FLC) population increases in number throughout fetal life, peaking around birth before gradually declining over the first 2 weeks of postnatal life[5]. It is generally thought that most adult Leydig cells (ALCs) arise de novo postnatally (i.e., FLCs generally do not directly give rise to ALCs to replace the FLC population); however, the idea that some FLCs persist in the adult testis has been proposed[5]. Recent lineage tracing studies have demonstrated that a subpopulation of FLCs is retained into adulthood, making up a small percentage (~5–20%) of total LCs in the adult testis, and a small number of FLCs may directly give rise to or transdifferentiate into ALCs[6,7]. ALCs have distinct morphological features and gene expression profiles compared to FLCs[8,9], and unlike FLCs, are able to produce testosterone on their own; mouse FLCs lack the enzymes critical for the final step in testosterone biosynthesis, such as HSD17B3, and thus only produce precursor androgens, such as androstenedione[10,11]. Therefore, fetal Sertoli cells are required to convert androstenedione from FLCs into testosterone.

Both fetal and adult LCs rarely divide and, therefore, rely on the differentiation of interstitial progenitors or stem cells to maintain a stable pool of mature LCs and to increase cell number during fetal and pubertal development[12–14]. Multiple putative progenitors for FLCs have been proposed, such as the coelomic epithelium (CE) and perivascular cells at the gonad–mesonephros border[15,16]. A recent single-cell RNA-seq study of Nr5a1-GFP-positive cells characterized the transcriptome of a putative interstitial progenitor population that gives rise to FLCs[17], and a recent lineage tracing study of gonadal progenitor populations described an early WT1-positive progenitor at E10.5 which gives rise to Sertoli cells, fetal Leydig and interstitial cells, and ALCs[18].

Another important event in fetal testicular development is testis-specific vascular remodeling to generate a coelomic arterial network[19,20], which supplies oxygen and nutrients to the growing gonad and is a conduit for export of testosterone. In addition to metabolic roles, blood vessels are proposed to play instructive roles during organogenesis[21]; for example, endothelial cell migration from the mesonephros and endothelial cell junctions are critical for establishing proper organ architecture during fetal testicular cord formation[22–24]. Studies have shown that VEGF signaling drives testicular vascular development and organ patterning, as blocking VEGF signaling severely impaired male-specific coelomic vessel formation and endothelial migration without grossly affecting Sertoli cell or Leydig cell specification[22,24].

Emerging evidence suggests that vasculature is a critical component of neural, mesenchymal, and hematopoietic stem cell niches[25–28]. Cross-talk between endothelial cells and tissue-specific stem cells within the niche is highly dynamic and is important for stem cell self-renewal and differentiation. Cell-to-cell interactions, such as those mediated by Notch signaling, regulate stem cell self-renewal and differentiation within perivascular niches[29–31].

In the fetal mouse testis, Notch signaling is required to maintain LC progenitors[32], mainly acting through the receptor NOTCH2[18]. Notch signaling restricts the differentiation of progenitor cells into mature FLCs to maintain the Leydig progenitor pool, but the identity of the cell types mediating critical Notch signaling is currently unknown. Since mature LCs are located close to the testicular vasculature, which is an arterial network[19], and arterial endothelial cells express Notch ligands[33,34], we hypothesized that endothelial cells are an integral component of Notch signaling in a perivascular niche for Leydig progenitors.

Here, we show that a multipotent progenitor population, which gives rise to several interstitial cell types, is located adjacent to the vasculature in the fetal testis. These perivascular progenitors express Nestin, an intermediate filament protein expressed in many stem cell populations[35]; Nestin-positive perivascular cells (e.g., vascular smooth muscle cells and pericytes) have been proposed as a progenitor population for ALCs[36,37], but Nestin's role in the fetal testis is unclear. Furthermore, we show that vasculature plays an essential instructive role in LC differentiation by maintaining perivascular Nestin-expressing progenitors. Additionally, we show that these progenitors are receiving an active Notch signal which is disrupted following vascular inhibition. Thus, Notch signals initiated by the vasculature are necessary for maintaining the balance between perivascular progenitor cells and differentiated LCs. Our findings support the idea that differentiation of fetal Leydig progenitors occurs by two different means, in which each has a unique cellular origin and distinct requirements for Notch signaling to maintain the progenitor population. Our studies uncover a new role for vasculature in testicular differentiation and shed light onto the origins of FLCs. Understanding the mechanisms underlying the maintenance and differentiation of interstitial progenitor cells is critical, as defects in LC differentiation in utero are an underlying etiology of gonadal dysfunction and represent a fetal origin of adult infertility and other reproductive conditions.

## Results

**Vasculature is dispensable for Sertoli cell differentiation.** While testis-specific vascular remodeling is important for establishing initial fetal testis cord architecture[22–24], it is not known whether vasculature is required for maintaining existing testis cord architecture or Sertoli cell differentiation, which could secondarily affect interstitial or Leydig cell differentiation. In order to determine vasculature's roles in these processes, we pharmacologically blocked VEGF signaling (via the VEGF receptor inhibitor VEGFR-TKI II) to disrupt vasculature in fetal testes ex vivo in a whole organ culture system. Quantitative real-time PCR (qPCR) analyses demonstrated that we effectively blocked vasculature, as expression of Cdh5 (also known as VE-cadherin), a vascular-endothelial-cell-specific gene, was significantly reduced in VEGFR-TKI-II-treated XY gonads compared to vehicle-treated controls (Fig. 1a). Immunofluorescence with anti-CDH5 antibody confirmed ablation of the vasculature in VEGFR-TKI-II-treated E12.5 fetal testes (Fig. 1b). However, the expression of Pou5f1 (also called Oct4), a germline-expressed gene, and Sox9, a Sertoli-cell-expressed gene, were not significantly different from controls, which is consistent with our previous findings[38] and shows that general testis health and differentiation are normal in vascular-depleted gonads. Furthermore, our analyses revealed that the expression of several Sertoli-expressed factors required for testicular development or LC differentiation was not affected by vascular disruption (Supplementary Fig. 1).

To determine whether disruption of vasculature induced hypoxia, and thus could potentially lead to deleterious secondary effects on testis development, we assessed the expression levels of Hif1a1 and Hif1a2, two isoforms of Hif1a (hypoxia inducible factor 1, alpha subunit), whose mRNA is stabilized and detectable

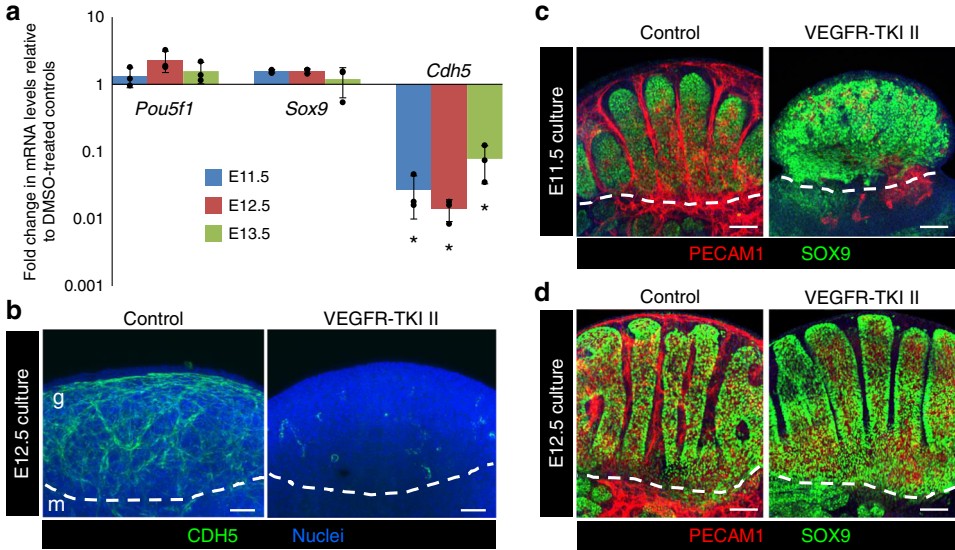

**Fig. 1** Vasculature is dispensable for Sertoli cell differentiation. **a** qPCR analysis representing fold change in expression of *Pou5f1* (also known as *Oct4*; germ cell gene), *Sox9* (Sertoli cell gene), and *Cdh5* (endothelial cell gene) in E11.5, E12.5, and E13.5 vascular-depleted fetal testes cultured for 48 h in the presence of VEGFR-TKI II (1.8 µg/µl) compared to DMSO-treated controls. Data are presented as the mean ± SD of 3 independent biological replicates (n = 3 litters, ≥5 gonads/litter). *P < 0.05 (two-tailed Student *t*-test). **b** Immunofluorescence images of E12.5 fetal testes cultured in the presence or absence (control; DMSO) of VEGFR-TKI II for 48 h. Relative to controls, vascular-depleted gonads showed a severe reduction of CDH5 (endothelial cell marker) immunostaining. **c, d** Immunofluorescence images of E11.5 (**c**) and E12.5 (**d**) fetal testes cultured in the presence or absence of VEGFR-TKI II stained for PECAM1 (germ/endothelial cell marker) and SOX9 (Sertoli cell marker). Dashed lines throughout indicate the border between the gonad (g) and mesonephros (m). Scale bars, 100 µm

only under hypoxic conditions[39]. The expression levels of *Hif1a1* and *Hif1a2* were not affected by vascular disruption in E12.5 XY gonads at normoxic conditions (Supplementary Fig. 2a). We also determined expression levels and subcellular localization of HIF1A protein via immunofluorescence. In contrast to gonads cultured in a hypoxic (1% oxygen) chamber (Supplementary Fig. 2b), treatment of cultured gonads with vascular inhibitor at normal oxygen levels did not result in changes of HIF1A protein levels or subcellular localization (Supplementary Fig. 2c), indicating that vascular disruption did not induce hypoxia. Additionally, immunofluorescence for cleaved Caspase 3 revealed that disruption of vasculature did not result in increased apoptosis (Supplementary Fig. 2d).

We next sought to determine if vasculature is essential for the initiation and maintenance of testis cord morphogenesis. Inhibition of VEGF signaling in cultured E11.5 fetal testes severely disrupted vascular remodeling and blocked testis cord formation (Fig. 1c), consistent with the previous studies[22,24,38]. However, inhibition of VEGF signaling in cultured E12.5 XY gonads still robustly blocked vasculature but did not affect existing testis cord structure (Fig. 1d).

**Undifferentiated perivascular cells express Nestin**. To characterize undifferentiated Leydig progenitors associated with the vasculature, we examined Nestin, a stem cell marker for various lineages[36,37,40,41] whose role in fetal testis development is poorly understood. Our previous transcriptome analyses of purified gonadal cell types showed that *Nestin* is expressed in interstitial mesenchymal cells and endothelial cells[42] (Supplementary Fig. 3a). Our immunofluorescence analyses of E13.5 testes revealed that Nestin protein was expressed in interstitial mesenchymal cells, but not within endothelial cells (Fig. 2a). Nestin-positive cells were specifically localized next to endothelial cells and displayed long processes that appeared to wrap around blood vessels (Fig. 2a). As expected, we observed that differentiated FLCs and Nestin-expressing cells were mutually

exclusive populations at both early (E13.5) and late (E18.5) stages of fetal development (Fig. 2b, c). Examining our previous cell-type-specific transcriptomic data[42], the pattern of *Nestin* expression was similar to markers of Leydig progenitors, such as *Lhx9*[15,32], *Arx*[13,43], and *Nr2f2* (also known as *Coup-tfii*)[44,45] (Supplementary Fig. 3a-d), and we found that Nestin-positive cells co-expressed ARX and NR2F2 (Supplementary Fig. 3e,f). Also consistent with the idea that Nestin-expressing cells are progenitor cells, we observed that Nestin-positive cells were often (but not always) positive for MKI67 (also known as Ki-67) (Fig. 2d), indicating that they were in active cell cycle. Use of the mitotic cell marker phospho-histone H3 (pHH3) confirmed that Nestin-positive cells were capable of proliferating (Fig. 2e). In contrast, differentiated FLCs, which did not express progenitor markers such as NR2F2 (Supplementary Fig. 3g), were very rarely MKI67-positive (Supplementary Fig. 3h) or pHH3-positive (Fig. 2e).

For later experiments to determine definitively whether *Nestin*-expressing cells are bona fide Leydig progenitors, we targeted *Nestin*-expressing cells using *Nestin*-Cre[46], and permanently lineage-traced them with a Cre-responsive *Rosa*-Tomato reporter. Our analyses showed that Tomato-expressing cells were localized adjacent to the vasculature in E11.5 and E13.0 gonads (Fig. 2f, g), thus recapitulating endogenous Nestin protein staining (see Fig. 2a). At earlier stages (E11.5), Tomato-positive cells were mostly limited to the mesonephros, in particular in the anterior end of the organ, whereas at later stages (E13.5), they were widespread mostly throughout the gonad.

**Nestin-positive mesonephric cells migrate into the gonad**. Given the proximity of *Nestin*-expressing cells to the vasculature and their mesonephric-specific presence in the E11.5 gonad (see Fig. 2f), we wanted to investigate further into the origin of these cells. To determine if these cells migrated into the gonad from the mesonephros, we performed gonad–mesonephros recombination assays using E12.0 wild-type control gonads and *Nestin*-Cre;

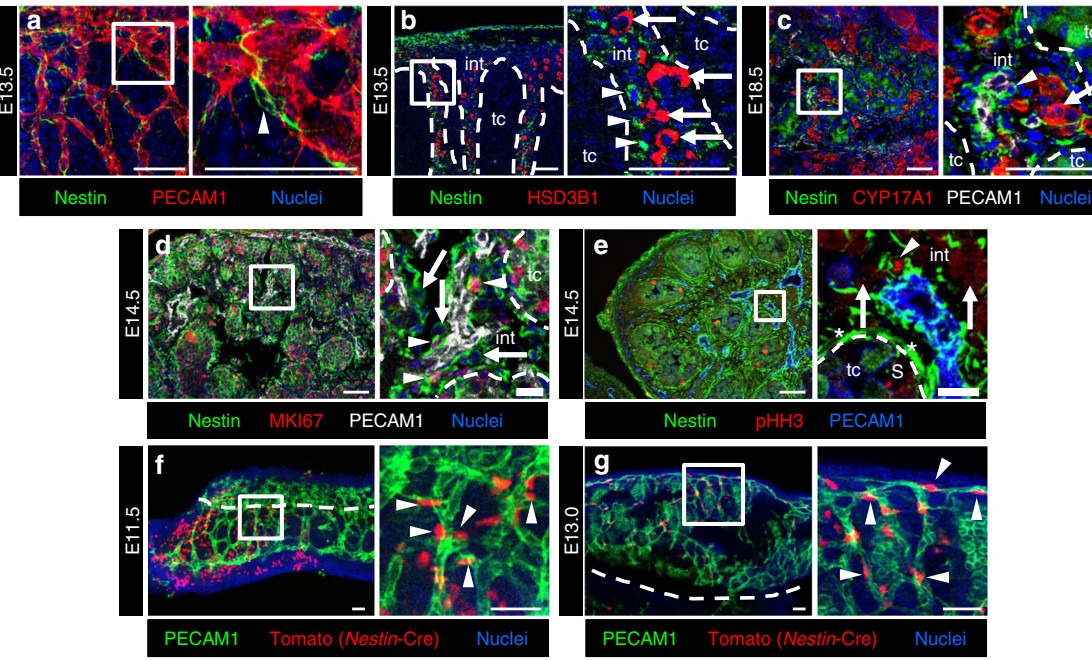

**Fig. 2** Undifferentiated perivascular cells express Nestin. Immunofluorescence images of E11.5 (**f**), E13.0 (**g**), E13.5 (**a**, **b**), E14.5 (**d**, **e**), and E18.5 (**c**) wild-type CD-1 (**a–e**) and *Nestin*-Cre; *Rosa*-Tomato (**f**, **g**) fetal testes. In E13.5 (**a**, **b**) and E18.5 (**c**) fetal testes, Nestin-positive cells (**a–c**, arrowheads) were localized close to the vasculature, as marked by PECAM1, and were mutually exclusive from differentiated Leydig cells, as marked by HSD3B1 (**b**, arrows) or CYP17A1 (**c**, arrow). Immunofluorescence images of E14.5 fetal testes (**d**, **e**) showed that Nestin-positive cells were MKI67-positive (**d**, arrowheads; marker for cells in active cell cycle) or pHH3-positive (**e**, arrowhead; marker for mitotic cells), while Leydig cells (**d**, **e**, arrows) were MKI67-negative and pHH3-negative. In (**e**), "S" marks mitotic Sertoli cell and asterisks denote the expression of Nestin in peritubular myoid cells. Immunofluorescence images of E11.5 (**f**) and E13.5 (**g**) lineage-traced *Nestin*-Cre; *Rosa*-Tomato testes showed that Tomato-positive cells (**f**, **g**, arrowheads) were located in close proximity to the vasculature. Dashed lines indicate the boundary of testis cords (tc) and interstitium (int) in (**b–e**) and the boundary between gonad and mesonephros in (**f**, **g**). Thin scale bars, 50 μm; thick scale bars, 10 μm

*Rosa*-Tomato mesonephroi (Fig. 3a). We detected many Tomato-positive cells adjacent to blood vessels throughout the wild-type gonad, some of which migrated through the entire fetal testis and resided near the surface coelomic artery (Fig. 3b). Additionally, we performed MitoTracker dye-labeling experiments of the E11.5 CE (Fig. 3c) to address whether *Nestin*-expressing cells were derived from the CE. We observed that *Nestin*-expressing cells, similar to endothelial cells (which are not derived from the CE), were not labeled by MitoTracker, even when located in a region of the gonad where adjacent cells were strongly dye-labeled (Fig. 3d–g). These results are consistent with *Nestin*-positive cells in the fetal testis arising from mesonephric cells that migrate into the gonad.

To address further whether these cells had a mesonephric or gonadal origin, we examined the expression of NR5A1 (also called SF1 or Ad4BP), which is specific to gonadal somatic cells and is not expressed in the mesonephros[47]. We found that Nestin-positive cells at E11.5, as well as *Nestin*-Cre; *Rosa*-Tomato-labeled cells at E13.5, were negative for NR5A1 (Supplementary Fig. 4a,b), consistent with a mesonephric, rather than a gonadal, origin. By E18.5, when some of these cells had attained a steroidogenic fate, they strongly expressed NR5A1 similarly to other LCs (Supplementary Fig. 4c). In contrast, Nestin-positive cells expressed WT1 (Supplementary Fig. 4d), which at E11.5 is expressed throughout the gonad and mesonephric mesenchyme[48,49].

**Nestin-positive cells give rise to FLCs and smooth muscle.** To determine if *Nestin*-positive cells are Leydig progenitors, we performed Tomato lineage tracing of *Nestin*-expressing cells in

*Nestin*-Cre; *Rosa*-Tomato fetal testes. Our immunofluorescence data revealed that at E13.5, differentiated FLCs were Tomato-negative (Fig. 4a; $n = 152$ cells), suggesting that the initial group of differentiated FLCs that first appear by E13.5 arise from a progenitor population other than *Nestin*-positive cells. However, by E18.5, approximately 33% of CYP17A1-positive cells expressed Tomato (Fig. 4b; $n = 800$ cells), indicating that a subpopulation of FLCs arise from *Nestin*-expressing progenitors.

To determine if *Nestin*-Cre-expressing cells give rise to other cell types in the fetal testis, we co-stained Tomato-labeled cells with: ACTA2 (also known as SMA or alpha-SMA), a marker for peritubular myoid cells (PMCs) and vascular smooth muscle; PECAM1, a marker for endothelial and germ cells; and SOX9, a marker for Sertoli cells. Our analyses revealed that virtually all ACTA2-positive cells, both peritubular and perivascular, co-expressed Tomato (Supplementary Fig. 5a). However, a lack of co-labeling with PECAM1 and SOX9 indicated that endothelial, germ, and Sertoli cell types did not arise from *Nestin*-expressing progenitors (Supplementary Fig. 5b,c).

To determine if *Nestin*-expressing cells give rise to ALCs and other cell types in the adult testis interstitium, we performed lineage tracing into adulthood in *Nestin*-Cre; *Rosa*-Tomato testes. Immunofluorescence analyses revealed that many HSD17B3 (a specific marker for ALCs)-positive cells expressed Tomato, indicating that a large subpopulation of ALCs arose from *Nestin*-expressing cells (Supplementary Fig. 6a); cell quantification analyses revealed that 47.2% of ALCs ($n = 644$ cells) were Tomato-positive. Some of the HSD17B3-negative FLCs retained in the adult testis[7] also expressed Tomato (Supplementary Fig. 6a). In addition, cells expressing ACTA2 and NG2 (a marker of pericytes, a perivascular cell type) cells co-expressed Tomato in

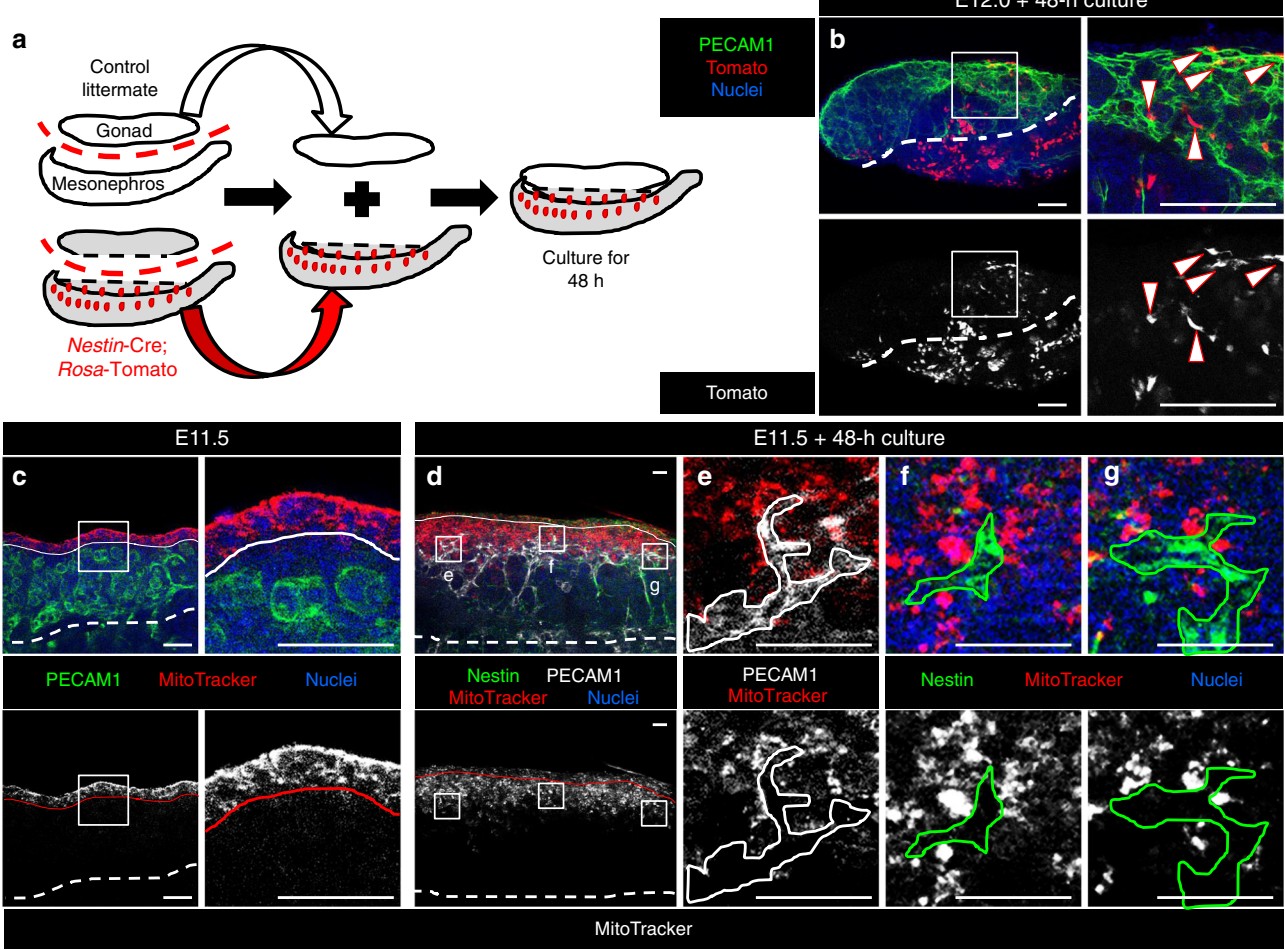

**Fig. 3** Nestin-positive mesonephric cells migrate into the gonad. **a** Schematic of a gonad–mesonephros recombination culture assay, in which control gonads are cultured adjacent to *Nestin*-Cre; *Rosa*-Tomato mesonephroi. **b**–**d** Immunofluorescence images of recombination culture (**b**) and MitoTracker-labeled gonads (**c**–**g**). **b** Tomato-positive cells (arrowheads) are observed in gonads after recombination culture for 48 h. The dashed line in (**b**) indicates the border between gonad and mesonephric components. **c** E11.5 gonad immediately after incubation with MitoTracker, showing that only the coelomic epithelium (above the solid line in (**c**)) was initially labeled. **d** After 48 h culture, MitoTracker label was observed below the coelomic epithelium (solid line in (**d**)) and deeper within the gonad. MitoTracker label was not observed within PECAM1-positive endothelial cells (white outline in (**e**)) nor within Nestin-positive cells (green outlines in (**f**, **g**)). Dashed lines in (**c**, **d**) indicate gonad–mesonephros border. Scale bars, 50 μm

the adult testis (Supplementary Fig. 6b,c), suggesting that *Nestin*-expressing cells gave rise to PMCs, vascular smooth muscle, and pericytes in the adult testis. However, the absence of Tomato expression in PECAM1- and SOX9-positive cells indicated that endothelial and Sertoli cells did not arise from *Nestin*-expressing progenitors in the adult testis (Supplementary Fig. 6b-d).

**Testicular perivascular cells are multipotent progenitors.** Immunofluorescence analyses revealed that Nestin is expressed in several interstitial cell types other than perivascular cells at later stages of fetal development (E14.5 and later; see Fig. 2d, e); therefore, it is possible that FLCs are derived from other *Nestin*-expressing cells that arise later during development. To address this possibility, we targeted perivascular cells specifically using tamoxifen-inducible *Nestin*-CreER mice[50,51]. By injecting pregnant mothers with 4-hydroxytamoxifen (4-OHT) at E12.25 and performing lineage tracing via *Rosa*-Tomato, we found that Tomato was specifically expressed in perivascular cells at E13.5 (Fig. 4c, d). Cytoplasmic Tomato protein clearly revealed how perivascular cells extended processes along and around blood vessels (Fig. 4c). *Nestin*-CreER at this stage did not target any non-perivascular mesenchymal cells in the interstitium, including

around the testis cords (Fig. 4d). As with *Nestin*-Cre, the initial cohort of FLCs at E13.5 was not labeled by Tomato (Fig. 4e).

By tracking perivascular cells until E18.5, we observed that they gave rise to FLCs (Fig. 4f). Interestingly, we also found that most peritubular myoid and vascular smooth muscle cells were labeled by Tomato at E18.5 (Fig. 4g). As in *Nestin*-Cre lineage tracing assays, *Nestin*-CreER-expressing perivascular cells did not give rise to Sertoli or endothelial cells (Supplementary Fig. 7a,b). Anti-NG2 antibody staining revealed that some Tomato-positive cells near blood vessels were pericytes (Supplementary Fig. 7c). Many Tomato-positive/Nestin-immunopositive cells remained adjacent to blood vessels in similar numbers to earlier stages (Fig. 4h), suggesting that these cells self-renewed rather than directly differentiated. These data indicate that perivascular Nestin-expressing cells are multipotent and give rise to multiple cell types in the fetal testis.

**Vascular disruption affects FLC progenitor differentiation.** To determine if vasculature is essential for regulating FLC differentiation, we disrupted vasculature in ex vivo whole gonad cultures at E12.5 (likely the developmental stage during which Leydig progenitors arise). Blocking vasculature at E12.5 resulted in a

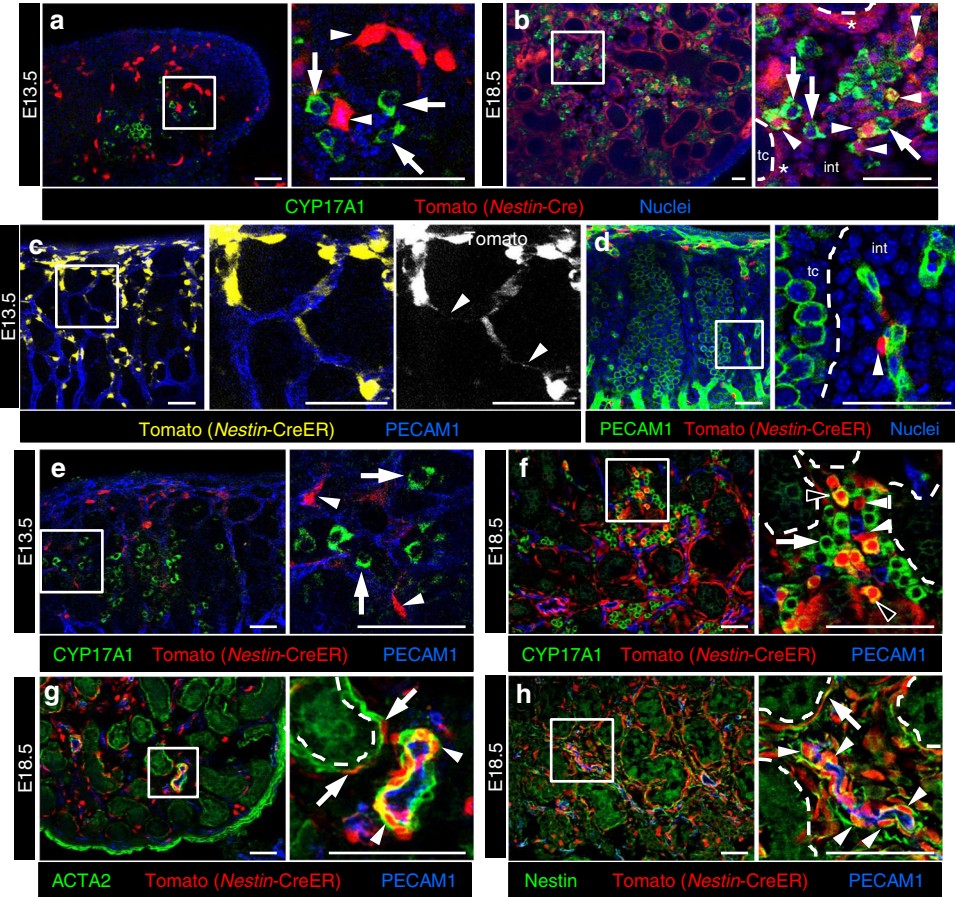

**Fig. 4** Perivascular Nestin-expressing cells are multipotent progenitors. Immunofluorescence images of E13.5 (**a**) and E18.5 (**b**) *Nestin*-Cre; *Rosa*-Tomato and E13.5 (**c–e**) and E18.5 (**f–h**) *Nestin*-CreER; *Rosa*-Tomato fetal testes exposed to tamoxifen at E12.25. At E13.5, the initial cohort of differentiated Leydig cells, as marked by CYP17A1 (**a**, **e**, arrows), were mutually exclusive from Tomato-lineage-traced *Nestin*-Cre-expressing cells (**a**, **e**, arrowheads). By E18.5, a subset of Tomato-labeled cells (**b**, arrowheads) expressed CYP17A1 (Tomato-negative FLCs are marked with arrows in (**b**, **e**, **f**)). Asterisks in (**b**) indicate Tomato expression in peritubular myoid cells and dashed lines in (**b**, **d**, **f–h**) indicate testis cord (tc)–interstitium (int) boundaries. In E13.5 *Nestin*-CreER; *Rosa*-Tomato fetal testes exposed to tamoxifen at E12.25 (**c–e**), initial Tomato-labeled cells were adjacent to the vasculature and extended long processes along the length of blood vessels (**c**, **d**, arrowheads), but no Tomato-positive cells were observed in the rest of the interstitial mesenchyme (**d**). By E18.5 (**f–h**), a subset of FLCs expressed Tomato (**f**, black arrowheads), in addition to peritubular myoid cells (**g**, arrows), vascular smooth muscle (**g**, arrowheads), and undifferentiated, Nestin-immunopositive perivascular cells (**f**, **h**, white arrowheads). Thin scale bars, 50 μm; thick scale bars, 10 μm

significant increase in the expression of FLC-specific steroidogenic genes (*Cyp11a1* and *Cyp17a1*), concomitant with significantly decreased expression of *Nestin* (Fig. 5a). Immunofluorescence analyses showed that more HSD3B1 (also known as 3β-HSD)-positive differentiated FLCs were produced at the expense of maintaining Nestin-positive cells following vascular disruption (Fig. 5b), indicating that changes in gene expression were due to changes in cell number of these populations.

To directly visualize the behavior of progenitor cells in the absence of vasculature, we disrupted vasculature in gonads with permanently Tomato-labeled *Nestin*-expressing cells. While in control testes, *Nestin*-expressing cells expanded in number and maintained their spindle-shaped morphology and lack of HSD3B1 expression, vascular disruption resulted in Tomato-positive cells changing their cellular morphology and expressing the differentiated LC marker HSD3B1 (Fig. 5c). These results suggest that Nestin-positive progenitors are prematurely or excessively differentiating into mature LCs in the absence of vasculature.

To confirm that these observed phenotypes were specifically due to disruption of VEGF-induced vascularization and not due to off-target effects of the small-molecule tyrosine kinase inhibitor VEGFR-TKI II, we performed ex vivo culture of fetal testes

treated with Aflibercept, also called VEGF-Trap, a potent peptide-based VEGF-specific inhibitor[52]. We found that Aflibercept-treated testes, as was observed in VEGFR-TKI-II treated testes, contained supernumerary LCs as compared to controls (Supplementary Fig. 8), suggesting that LC differentiation is specifically regulated by VEGF-dependent testicular vascularization.

**Nestin progenitors receive active Notch signal**. To determine if perivascular Nestin-positive progenitors undergo active Notch signaling, we performed immunofluorescence analyses of E13.5 *CBF*: H2B-Venus Notch reporter gonads. Many (approximately 30%) of Nestin-positive cells were undergoing active Notch signaling (Fig. 6a), along with a subset of endothelial cells (Fig. 6b). However, the first cohort of FLCs at E13.5 was virtually absent of active Notch signaling (Fig. 6c), as only 1.1% of FLCs expressed Venus ($n = 264$ cells; Fig. 6f), whereas approximately 40% of differentiated FLCs at E18.5 expressed Venus ($n = 1574$ cells; Fig. 6d–f), most of which weakly expressed Venus relative to vascular and perivascular cells. Given the perdurance of H2B-Venus within cells, this mouse line has been used as a short-term lineage tracer[53], so these data suggest that later-born FLCs were derived from progenitor cells that previously underwent (and likely were previously maintained by) active Notch signaling.

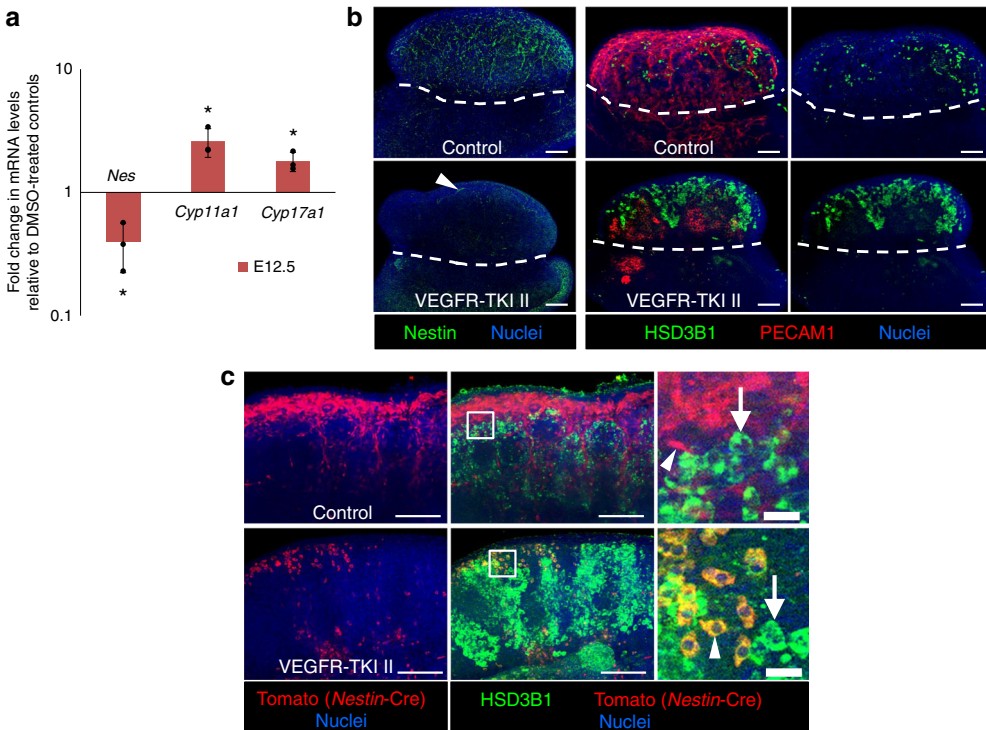

**Fig. 5** Vascular disruption induces excessive differentiation of Nestin-positive cells. **a** qPCR analysis showing fold change in expression of *Nes* (*Nestin*; interstitial progenitor gene), *Cyp11a1*, and *Cyp17a1* (differentiated FLC genes) in E12.5 fetal testes cultured for 48 h in the presence of VEGFR-TKI II (1.8 µg/ µl) relative to DMSO-treated controls. Data are presented as the mean ± SD of 3 independent biological replicates (*n* = 3 litters, ≥5 gonads/litter). *P < 0.05 (two-tailed Student *t*-test). **b** Immunofluorescence images of E12.5 fetal testes cultured for 48 h in the absence (top row, DMSO Control) or presence (bottom row) of VEGFR-TKI II (1.8 µg/µl). Following vascular depletion, there was an increase in the number of HSD3B1-positive cells and a decrease in the number of Nestin-positive progenitors compared to DMSO-treated control gonads. Arrowhead in (**b**) shows some remnant Nestin expression in peritubular cells of vascular-depleted gonads. **c** Immunofluorescence images of E12.5 *Nestin*-Cre; *Rosa*-Tomato testes cultured for 48 h in the absence (DMSO Control, top row) or presence of VEGFR-TKI II (1.8 µg/µl, bottom row). In DMSO-treated control gonads, Tomato-positive cells were spindle-shaped (top row, arrowhead) and mutually exclusive from HSD3B1-positive Leydig cells (arrows), while in vascular-depleted gonads, Tomato-positive cells started to express HSD3B1 (bottom row, arrowhead) and are round in shape, indicating their (premature) differentiation into mature FLCs following vascular depletion. Thin scale bars, 100 µm, thick scale bars, 10 µm

We next addressed the issue of which receptors and ligands are involved in active Notch signaling in the fetal testis interstitium. Our previous transcriptome analyses indicated that *Notch2* and *Notch3* are the only receptor-encoding genes expressed in the interstitial mesenchyme of the fetal testis[42,54]; however, previous studies have implicated the involvement of *Notch2*[18], but not *Notch3*[55] in FLC differentiation. To verify directly that NOTCH2-mediated signaling occurs in the fetal testis, we used a *Notch2-ICD*-Cre mouse line, in which active NOTCH2 signaling induces Cre activity[56]. We permanently lineage-labeled NOTCH2-active cells using a *Rosa*-YFP reporter, and observed that perivascular, Nestin-positive cells were labeled with YFP in the E18.5 fetal testis (Supplementary Fig. 9a, b). We also observed that a few differentiated LCs were YFP-positive (Supplementary Fig. 9c), suggesting that these cells arose from progenitors previously maintained by NOTCH2. The number of cells labeled by YFP was lower than anticipated, likely due to this Cre line being a low-sensitivity Cre line[56].

Our previous transcriptome analyses[42,54] suggested that the Notch ligand most likely expressed by the vasculature in the fetal testis was *delta like canonical Notch ligand 4* (*Dll4*). Immuno-fluorescence revealed that DLL4 expression was specific to endothelial cells within the fetal testis, but there was minimal expression in the mesonephric vasculature (Supplementary Fig. 9d). We also confirmed that DLL4-positive endothelial cells were adjacent to NOTCH2-positive interstitial mesenchymal cells and Nestin-positive cells (Supplementary Fig. 9e,f).

**Vascular inhibition disrupts testicular Notch signaling**. Following our *CBF*:H2B-Venus results, we wanted to determine if vasculature is required for active Notch signaling in the testis interstitium. First, we assessed the expression of Notch target genes and observed that *Hey1* and *HeyL* were significantly decreased upon vascular disruption (Fig. 7a). *HeyL* is of particular interest because it is interstitial-mesenchyme-specific and has a similar expression pattern to Leydig progenitor-specific genes (Fig. 7b; see also Supplementary Fig. 3a-d). As an additional way to test the role of the vasculature in driving active Notch signaling, we disrupted vasculature ex vivo in E12.5 *CBF*:H2B-Venus fetal testes and saw that vascular inhibition led to loss of active Notch signal specifically in the interstitium (Fig. 7c). Sertoli cells, which we have previously shown undergo active Notch signaling[54,57], retained Venus expression, indicating that loss of vasculature did not affect active Notch signaling in Sertoli cells.

**Blocking Notch signaling induces loss of Nestin progenitors**. To assess if Notch signaling is essential for the maintenance of perivascular progenitor cells, we inhibited Notch signaling using the gamma-secretase inhibitor DAPT. Treatment of fetal testes with 100 µM DAPT effectively blocked Notch signaling, as expression of all Notch target genes (*Hes1*, *Hes5*, *HeyL*, and *Hey1*) tested was significantly downregulated (Fig. 8a). Additionally, blocking Notch signaling had no effect on *Sox9* expression

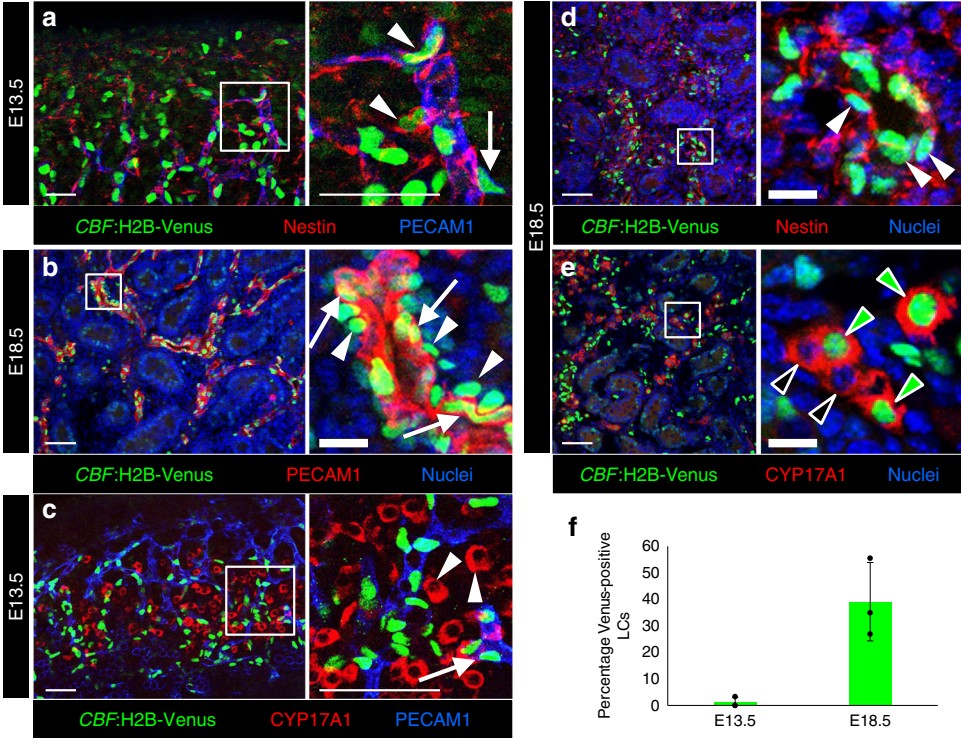

**Fig. 6** Active Notch signaling in the fetal testis. **a–e** Immunofluorescence images of E13.5 (**a**, **c**) and E18.5 (**b**, **d**, **e**) fetal testes from *CBF*:H2B-Venus embryos. **a**, **b** A subset of PECAM1-positive endothelial cells (arrows) and Nestin-positive cells adjacent to the vasculature (arrowheads) were Venus-positive. **c** The initial cohort of FLCs (CYP17A1-positive; arrowheads) was negative for Venus, while vasculature-associated cells (arrow) expressed Venus. **d**, **e** At E18.5, Nestin-positive cells expressed Venus (**d**, arrowheads), while differentiated FLCs were either Venus-positive (**e**, green arrowheads) or Venus-negative (**e**, black arrowheads). Thin scale bars, 50 μm; thick scale bars, 10 μm. **f** Graph showing the percentage of CYP17A1-positive cells expressing Venus at E13.5 ($n = 264$ cells from 3 independent gonads) and E18.5 ($n = 1574$ cells from 3 independent gonads). Data are shown as mean ± SD

(Fig. 8b), and immunofluorescence analyses did not reveal any noticeable effect on the vasculature in DAPT-treated testes (Fig. 8c).

Disrupting Notch signaling also resulted in an increase in the expression of *Cyp11a1* and *Cyp17a1*, concomitant with a significant decrease in *Nestin* expression (Fig. 8b). Immunofluorescence analyses confirmed that Notch inhibition by DAPT induced supernumerary FLCs and a severe loss of Nestin-positive cells (Fig. 8c). These results indicate that Notch signaling is responsible for regulating the number of FLCs by maintaining Nestin-positive progenitor cells in the fetal testis.

## Discussion

Growing evidence suggests that vasculature plays instructive roles during organogenesis[21]. In the fetal testis, vasculature is critical for the initial formation and patterning of testis cords[22–24], structures that give rise to the seminiferous tubules in the adult testis. Whether vasculature is required to maintain testis cord architecture was not known. In this study, we demonstrate that vasculature is required for the initial formation, but not for the maintenance, of fetal testis cord architecture. Our data indicate that vasculature plays an instructive role in testis morphogenesis, independent of any secondary effects on Sertoli cell differentiation or hypoxia after vascular disruption.

Perivascular cells (including pericytes and vascular smooth muscle) can act as multipotent progenitors capable of differentiation into bone, cartilage, and adipose cells[25]. Multipotent perivascular cells express Nestin, an intermediate filament protein found in the progenitor cells of neural and non-neural tissue[58], including in the adult testis[36]. Characterization of Nestin-positive cells in the adult rat testis showed that they were putative

progenitors for ALCs, exhibited stem-cell-like properties, and were able to differentiate into multiple lineages[36,37].

Whether fetal vasculature-associated Nestin-positive progenitors differentiate into FLCs was not known. Here, we demonstrate that Nestin-positive cells are bona fide fetal Leydig progenitors in vivo, and are closely associated with the vasculature at multiple developmental stages, consistent with studies in adult rat testis[36,37]. Our data also suggests that *Nestin*-lineage-traced cells migrated from the mesonephros into the gonad, likely along with endothelial and mesenchymal cells during testis vascular remodeling[15,19,20,23].

Lineage tracing of *Nestin*-expressing progenitors revealed that LCs at E13.5 were not among *Nestin*-lineage-labeled cells. Hence, the first cohort of differentiated LCs comes from a progenitor population other than *Nestin*-positive cells. We have previously proposed that multiple different sources exist for the origin of FLCs[15], so *Nestin*-positive cells likely represent one of these major sources. At E18.5, a large subset (~33%) of FLCs were derived from *Nestin*-expressing cells. The presence of 33% of *Nestin*-progenitor-derived LCs, as opposed to 100%, reinforces the likelihood of another progenitor population, which likely arises from proliferation of the CE; we have previously shown via dye-labeling assays that coelomic epithelial cells can give rise to LCs[15]. It is also possible that the efficiency of *Nestin*-Cre-mediated recombination is suboptimal, leading to fewer Tomato-labeled progenitors that differentiate into LCs; however, our co-localization analyses examining Tomato and Nestin protein expression indicate that both Cre and CreER activity are at least 90% effective (see Figs. 2 and 4h).

Our characterization of *Nestin* progenitors showed that they expressed other Leydig progenitor markers, such as ARX and

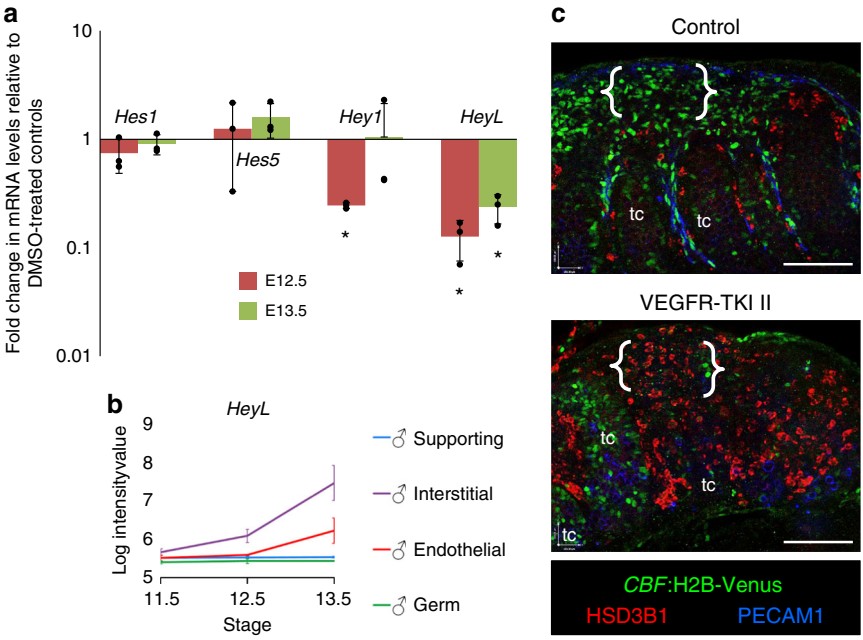

**Fig. 7** Vasculature is required for interstitial Notch signaling. **a** qPCR analysis showing expression of Notch target genes (*Hes1*, *Hes5*, *Hey1*, and *HeyL*) from E12.5 and E13.5 fetal testes cultured with VEGFR-TKI II (1.8 µg/µl) for 48 h relative to their respective vehicle (DMSO)-treated controls. Data are presented as the mean ± SD of 3 independent biological replicates (n = 3 litters, ≥5 gonads/litter). *P < 0.05 (two-tailed Student t-test). **b** Gene expression plot of *HeyL* expression generated from gonad lineage-specific microarray data[42], where cell lineages (supporting Sertoli cells, interstitial cells, endothelial cells, and germ cells) were independently plotted in different colors. Plot contains data for XY samples at stages E11.5, E12.5, and E13.5 for each cell type. In general, expression values below 6 are considered background levels of expression. For more information about this data set, see Jameson et al.[42]. **c** Immunofluorescence images of E12.5 *CBF*:H2B-Venus XY gonads cultured in DMSO (Control, top) or 1.8 µg/µl VEGFR-TKI II (bottom) for 48 h. Disruption of vasculature resulted in loss of active Notch signal from the interstitium (Venus-positive cells; see brackets) and resulted in an increased number of differentiated Leydig cells (HSD3B1-positive); "tc" denotes testis cords

NR2F2, and were in active cell cycle and proliferated, consistent with the idea that they are bona fide Leydig progenitors. In addition to giving rise to LCs, lineage-labeled *Nestin*-positive cells also gave rise to PMCs, vascular smooth muscle, and pericytes in the E18.5 fetal testis. Our lineage tracing analyses using *Nestin*-CreER showed that PMCs are largely derived from perivascular progenitors rather than from a general mesenchymal precursor, which is a significant finding since the origins of PMCs are currently unclear[59,60]. We also found that a subset of perivascular cells were pericytes, suggesting that pericytes are a multipotent progenitor population in the fetal testis similar to what has been described for other tissues[25].

Whether *Nestin*-expressing cells are bona fide ALC precursors in vivo during normal development was also an open question. Our adult lineage tracing analyses revealed that a significant subset of ALCs (~50%) were Tomato-positive (as well as a subset of retained FLCs in the adult testis), which definitively demonstrates in vivo that *Nestin*-expressing cells are ALC progenitors during normal development. This finding is consistent with previous studies that used in vitro, transplantation, and injury models[36,37] to propose that *Nestin*-expressing cells are ALC progenitors. Other cell types in the adult testis, such as PMCs, vascular smooth muscle, and pericytes, were also derived from *Nestin*-positive cells, further demonstrating their multipotent differentiation capability.

Dye labeling experiments showed that cells derived from the CE gave rise to both Sertoli cells and Leydig cells in the fetal gonad, thereby representing an early common progenitor for Sertoli and interstitial cells in the bipotential gonad[15,16]. Consistent with this idea, a recent lineage tracing study showed that a single WT1-positive somatic progenitor population at E10.5 gave rise to fetal Sertoli, interstitial (non-steroidogenic), and Leydig

cells, as well as ALCs[18]. Additional studies demonstrated that *Wt1* is required for preventing transdifferentiation of Sertoli cells into Leydig cells, via the repression of *Nr5a1* expression[61,62], consistent with Sertoli and Leydig cells having a common progenitor. Our analyses showed that early Nestin-positive cells are WT1-positive. Thus, we have identified a unique Nestin-positive, WT1-positive interstitial progenitor population with a mesonephric origin. A recent single-cell transcriptome study of fetal gonads described a single, distinct progenitor population for interstitial cells[17]. However, that study purified gonadal cells on the basis of *Nr5a1*-GFP transgene expression, and our data suggests that *Nestin*-expressing progenitors do not express NR5A1 (Supplementary Fig. 4); therefore, this cell type was likely not included in their analyses.

Vasculature has emerged as a critical component of many stem cell niches, playing instructive roles in the perivascular niche microenvironment. Our data shows that vasculature is important in maintaining the interstitial progenitor population that gives rise to differentiated LCs. In the absence of vasculature, lineage-labeled Nestin-positive cells express steroidogenic enzymes and prematurely differentiate into LCs. These findings indicate that Nestin progenitors are responsive to vascular cues, which, ultimately, are essential for regulating (and inhibiting) Leydig differentiation. It is important to note that in vascular-disrupted gonads in which Nestin-expressing cells are lineage-traced with Tomato, many of the supernumerary LCs are Tomato-negative, suggesting that progenitors other than Nestin-expressing cells also prematurely differentiate into mature LCs following vascular inhibition.

Within a perivascular niche, cell–cell interaction mediated by Notch signaling maintains neural and hematopoietic stem cell populations[29–31]. Genes encoding Notch ligands, such as *Dll4*,

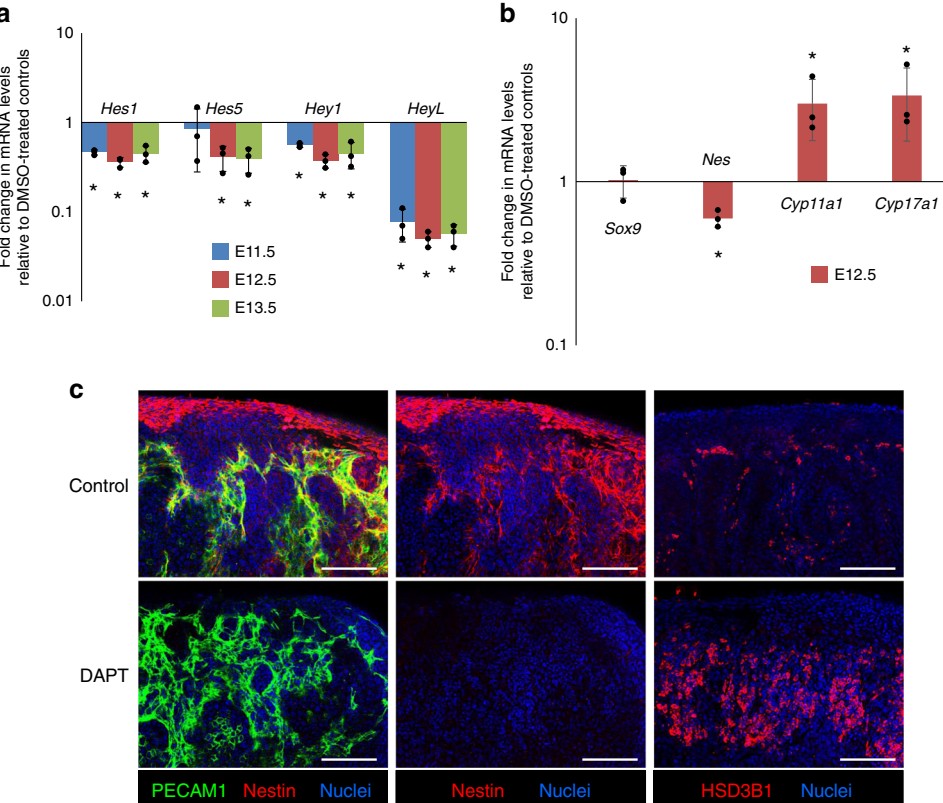

**Fig. 8** Disrupting Notch signaling induces loss of *Nestin*-positive progenitors. **a**, **b** E11.5, E12.5, and E13.5 fetal testes were cultured with 100 µM DAPT for 48 h and fold change in gene expression was calculated relative to their respective vehicle (DMSO)-treated control testes. **a** qPCR analysis showing fold change in expression of Notch target genes expressed in the fetal testis (*Hes1*, *Hes5*, *Hey1*, and *HeyL*). **b** qPCR analysis showing fold change in expression of *Sox9* (Sertoli cells), *Cdh5* (endothelial cells), *Nes* (interstitial progenitors), and *Cyp11a1* and *Cyp17a1* (differentiated Leydig cells) after 48-h DAPT treatment of cultured E12.5 fetal testes relative to control cultured testes. Data are presented as the mean ± SD of 3 different biological replicates ($n = 3$ litters, $\geq 5$ gonads/litter). *$P < 0.05$ (two-tailed Student *t*-test). **c** Immunofluorescence of E12.5 fetal testes cultured in DMSO (Control) or 100 µM DAPT for 48 h, showing loss of Nestin-positive cells and increase in HSD3B1-positive FLCs in DAPT-treated samples

*Jag1*, and *Jag2*, are expressed in endothelial cells, including in the fetal testis[19,42,54], making it reasonable to hypothesize that endothelial cells are the predominant cell type providing Notch ligands within testicular perivascular niches. Our data using *CBF*: H2B-Venus reporter mice showed that Nestin-positive cells adjacent to blood vessels receive active Notch signal; furthermore, active Notch signal in the interstitium is absent upon vascular disruption, consistent with the idea of endothelial cells participating in Notch signaling with perivascular cells.

Our analyses showed that the first cohort of differentiated FLCs at E13.5 lack any residual Notch reporter expression, indicating that when LC specification first begins, interstitial gonadal cells lack active Notch signal and progenitor cells readily or spontaneously differentiate into mature LCs. At E18.5, about 40% LCs were Venus-positive, indicating that their progenitors received, and were previously maintained by, active Notch signaling. Our previous transcriptome and immunofluorescence analyses demonstrated that NOTCH2 is expressed in the interstitium and is likely involved in activating Notch signaling in Nestin-positive progenitors[42,54]. Consistent with these findings, a recent study showed that *Notch2* is essential for maintaining Leydig progenitors[18], and our data using a NOTCH2-responsive Cre mouse line is consistent with NOTCH2-mediated signaling in perivascular cells. NOTCH3 is the other Notch receptor expressed in the interstitium[42,54]. Hence, it is possible that Notch activation through NOTCH3 also regulates Leydig progenitor maintenance and differentiation in the fetal testis; further investigation of *Notch3* function may prove informative. However, previous

genetic analyses reported no developmental or reproductive defects in *Notch3*-mutant mice[55].

Our findings demonstrate that active Notch signal (via the receptor NOTCH2) in perivascular mesenchymal cells, driven by endothelial Notch ligands (likely DLL4), maintains the progenitor pool either through self-renewal or maintenance of an undifferentiated state; our MKI67 and pHH3 analyses suggest that perivascular cells are in active cell cycle and can potentially expand and/or self-renew. As fetal development progresses, the need for LCs rises and undifferentiated progenitor cells become competent to differentiate and have low or absent Notch signaling activity. Our results led us to propose a new idea regarding LC development, in which the differentiation of FLCs occurs via two distinct processes: epithelial-derived or perivascular-derived (Fig. 9). The first cohort of LCs arises from progenitors that are regulated independently of Notch; this early cohort is likely derived directly from the CE, which we and others have previously shown gives rise to interstitial and Leydig cells when the division of the epithelium occurs at or after E11.5[15,15]. The second process occurs via the differentiation of Nestin-positive perivascular progenitors that are maintained in an undifferentiated state by active Notch signaling and differentiate upon loss of active Notch signal, perhaps due to asymmetric cell division and subsequent loss of contact with endothelial cells. These cells are likely migrating mesenchymal cells from the gonad/mesonephros border, which we have shown can also give rise to FLCs[15] and which others have shown give rise to ovarian steroidogenic theca cells later in development[63], and mostly contribute to later-born LCs.

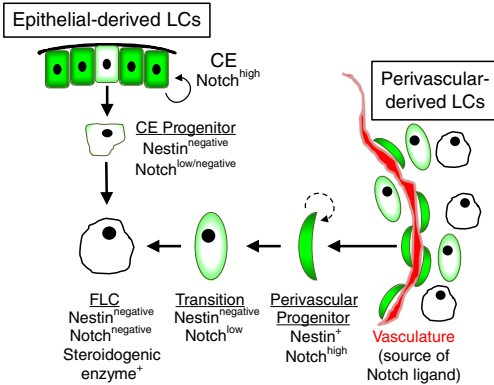

**Fig. 9** Vasculature-dependent Notch activation regulates FLC progenitor differentiation. Our data support a model in which there are 2 distinct means of Leydig cell differentiation: epithelial-derived and perivascular-derived. The first cohort of Leydig progenitors is derived from the coelomic epithelium (CE), which initially receives Notch signal early during gonadogenesis (but not in cells that ingress into the gonad and differentiate into committed lineages such as Sertoli cells; as shown in our recent work[67]), but during proliferation and ingression into the gonad loses active Notch signal and differentiates directly into Leydig cells independently of Notch-maintained progenitors. Later cohorts of Leydig cells are derived from mesenchymal perivascular progenitors that co-migrate with endothelial cells from the mesonephros into the testis (or alternatively could be originally derived from the CE). Endothelial-derived Notch ligands activate Notch signaling to maintain Nestin-positive perivascular cells in an undifferentiated state and in active cell cycle with the potential to expand or self-renew. Some Nestin-positive progenitors, which initially exhibit active Notch signaling, become competent to differentiate into mature Leydig cells (e.g., express steroidogenic enzymes) following a loss of active Notch signal

Consistent with the idea that there are temporally unique requirements for Notch signaling in FLC differentiation from their progenitors, a recent study reported that the effects of *Notch2* deletion on FLC differentiation were significantly more severe at later stages (newborn) as compared to earlier stages (E13.5)[18].

Our observations support the idea of a Notch-mediated crosstalk between endothelial cells and perivascular progenitors that regulates LC differentiation. Disruption or loss of active Notch signaling in progenitor cells induces precocious differentiation into mature LCs and premature exhaustion of the progenitor pool. The subsequent deficit of mature LCs would adversely affect testosterone production and lead to harmful effects on sexual differentiation and fertility. In all, our data has provided new insights into FLC development by uncovering an instructive role for blood vessels in maintaining interstitial progenitor cells in the fetal testis.

## Methods

**Mice**. C57BL/6J mice (JAX stock #000664) or outbred CD-1 mice (Charles River) were used for wild-type expression studies. *Rosa*-Tomato mice (also called Ai14), a Cre-responsive tdTomato reporter line (B6.Cg-*Gt(ROSA)26Sor*tm14(CAG-tdTomato)Hze/J), were obtained from The Jackson Laboratory (JAX stock #007914) and are maintained on a C57BL/6J (B6) background. *Nestin*-Cre mice[46] (Tg(Nes-cre)1Kln/J), on a B6 background, were a gift from Dr. Richard Lang (Division of Pediatric Ophthalmology, Cincinnati Children's Hospital Medical Center; JAX stock #003771). *Nestin*-CreER mice[50,51] (Tg(Nes-cre/ERT2,-ALPP)1Sbk), on a mixed B6/FVB/CD-1 genetic background, were obtained from Dr. Masato Nakafuku (Division of Developmental Biology, Cincinnati Children's Hospital Medical Center). *CBF*:H2B-Venus mice[53] (Tg(Cp-HIST1H2BB/Venus)47Hadj/J)), on a mixed FVB/NJ and CD-1 background, were obtained from The Jackson Laboratory (JAX stock #020942). *Notch2-ICD*-Cre mice (*Notch2*tm2(cre)Rko; also called N2::Cre[LO])[56,64], a low-sensitivity Cre line in which the intracellular domain (ICD) of NOTCH2 was replaced with Cre, were used to lineage-trace cells that underwent

NOTCH2 signaling via a Cre-responsive *Rosa*-YFP reporter strain (*Gt(ROSA)26Sor*tm3(CAG-EYFP)Hze/J; also called Ai3, JAX stock #006148); these strains are maintained on a mixed background and were provided by Dr. Raphael Kopan via Dr. Joo-Seop Park (Divisions of Developmental Biology and Urology, Cincinnati Children's Hospital Medical Center). Mice were housed in accordance with National Institutes of Health guidelines, and experimental protocols were approved by the Institutional Animal Care and Use Committee (IACUC) of Cincinnati Children's Hospital Medical Center (animal experimental protocol numbers IACUC2013-0241 and IACUC2018-0027).

**Lineage tracing of *Nestin*-expressing cells**. For *Nestin*-Cre experiments, Cre-positive male mice were crossed with *Rosa*-Tomato females. For *Nestin*-CreER experiments, CreER-positive males were crossed with *Rosa*-Tomato females; pregnant females were injected intraperitoneally at E12.25 with 75 μg/g 4-hydroxytamoxifen (4-OHT; Sigma-Aldrich #H6278) dissolved in corn oil, along with 37.5 μg/g progesterone (Sigma-Aldrich #P0130) subcutaneously (to help maintain pregnancy). For E18.5 experiments, an additional progesterone injection was given at E14.5 to help maintain pregnancy further after 4-OHT administration. Pregnant females were euthanized at the required stage and gonads from embryos were dissected and processed for immunofluorescence or organ culture. For lineage tracing into adulthood, testes from *Nestin*-Cre; *Rosa*-Tomato males were analyzed at 60 days old (P60).

**Whole-organ gonad droplet culture**. Whole-gonad droplet cultures were performed on E11.5–E13.5 fetal testes from CD-1, *Nestin*-Cre; *Rosa*-Tomato, or *CBF*:H2B-Venus embryos. One XY gonad of each pair was cultured in a 30-μL droplet of complete medium (DMEM/10% fetal bovine serum [FBS]/1% penicillin–streptomycin) containing vehicle (DMSO), while the contralateral gonad was cultured in a 30-μL droplet of complete medium containing either 1.8 μg/mL VEGF Receptor Tyrosine Kinase Inhibitor II (VEGFR-TKI II; Calbiochem/EMD Millipore #676481-5MG) for vascular depletion experiments[38] or 100 μM DAPT for blocking of γ-secretase activity[32]. For Aflibercept/VEGF-Trap experiments, 38 μg/μL Aflibercept (Regeneron) was delivered to the vasculature via microinjection into beating E11.5 hearts, as described in our previous reports[15,24]. This reagent is a soluble decoy receptor in which the first three Ig domains of human VEGFR1 are fused to the constant region (Fc) of human IgG1 and is a potent VEGF inhibitor that binds to VEGFA with picomolar affinity[52]. Following culture for 48 h, the gonad–mesonephros complexes were either fixed for immunofluorescence or the gonads were separated from the mesonephros and pooled for RNA extraction and qPCR analysis. Sex of E11.5 embryos was determined via a X- and Y-chromosome-specific PCR method[65].

**Immunofluorescence**. Tissues were dissected in phosphate-buffered saline (PBS) and fixed in 4% paraformaldehyde (PFA) with 0.1% Triton X-100 overnight at 4 °C. Whole-mount immunofluorescence was performed on fetal gonads at stages E13.5 and younger. After several washes in PBS + 0.1% Triton X-100 (PBTx), samples were incubated in blocking solution (PBTx + 10% FBS + 10% bovine serum albumin [BSA]) for 1–2 h at room temperature. Primary antibodies were diluted in blocking solution and applied to samples rocking overnight at 4 °C. After several washes in washing solution (PBTx + 1% FBS + 3% BSA), fluorescent secondary antibodies were applied for 4–5 h rocking at room temperature or overnight at 4 °C. After several washes in PBTx, samples were mounted on slides in Fluoromount-G (SouthernBiotech).

For fetal testes at stages E14.5 and older, cryosections were performed. Dissections and fixation were performed as above. After several washes in PBTx, samples were processed through a sucrose:PBS gradient (10, 15, 20% sucrose) before an overnight incubation in a 1:1 mixture of 20% sucrose and OCT medium (Sakura) rocking at 4 °C. Samples were embedded in OCT medium at −80 °C prior to cryosectioning. Cryosections were then stained and mounted as above, except secondary antibodies were applied for only 1 h.

For adult testes, the tunica of the testis was gently punctured 10–12 times with a 27-gauge needle before being fixed in 4% PFA + 0.1% Triton X-100 overnight at 4 °C. Samples were then cut in half transversely with a scalpel and fixed further for 1 h in PFA + 0.1% Triton X-100 at 4 °C to ensure fixative was able to access the center of the tissue. After several washes in PBTx, samples were processed through sucrose as above for fetal cryosectioning before being embedded in OCT medium. For HSD17B3 staining, sections were subjected to antigen retrieval by autoclaving in 10 mM sodium citrate solution (pH 6.0) for 3 min.

Primary antibodies and dilutions used in this study are listed in Supplementary Table 1. *Rosa*-Tomato and *CBF*:H2B-Venus expression were detected via endogenous fluorescence, except for HSD17B3 experiments, in which antigen retrieval weakened tdTomato fluorescence and necessitated use of anti-RFP (anti-tdTomato) antibody, and *Notch2*-ICD-Cre lineage-tracing experiments, in which anti-GFP antibody was used to amplify YFP signal. Alexa-488-, Alexa-555-, and Alexa-647-conjugated secondary antibodies (Molecular Probes/Life Technologies/Thermo Fisher) and Dy-Lite 488 donkey anti-chicken and Cy3 donkey or goat anti-rabbit/rat secondary antibodies (Jackson ImmunoResearch) were used at 1:500. Nuclei were stained with 2 μg/mL Hoechst 33342 (#H1399, Molecular Probes/Life Technologies/Thermo Fisher), and Hoechst staining is labeled as

"Nuclei" in all figures. Images were taken on a Nikon Eclipse TE2000 microscope (Nikon, Tokyo, Japan) equipped with an Opti-Grid structured illumination imaging system running Volocity software (PerkinElmer, Waltham, MA, USA) or on a Nikon A1 Inverted Confocal Microscope (Nikon, Tokyo, Japan). At least two independent experiments were performed and within each experiment multiple XY gonads ($n = 4$–8) were used.

**Whole-gonad hypoxia culture**. For hypoxia experiments, one E12.5 CD-1 gonad of each pair was cultured via droplet in a normoxic chamber incubator (20% $O_2$, 5% $CO_2$, 37 °C) and the contralateral gonad was cultured via droplet in a hypoxic chamber incubator (1% $O_2$, 5% $CO_2$, 37 °C) for 24 h. Following culture, the gonads were immediately and directly placed into 4% PFA with 0.1% Triton X-100 within the incubator and then moved to 4 °C for overnight fixation. Gonads were then subjected to immunofluorescence as described above.

**MitoTracker coelomic epithelial labeling**. E11.5 CD-1 gonads were incubated in 1 mM MitoTracker Orange CMTMRos (Thermo Fisher Scientific/Invitrogen #M7510) in PBS for 30 min at 37 °C to label the surface CE. Following incubation with MitoTracker, gonads were washed several times in culture media. Gonads were then cultured for 48 h in 1.5% agar wells and processed for whole-mount immunofluorescence as described above. As a control, several gonads in each experiment were removed and fixed immediately after labeling and washing with culture media to ensure that only the surface coelomic epithelial cells were initially labeled.

**Gonad–mesonephros recombination culture**. Gonad–mesonephros complexes were dissected from E12.0 embryos and were sexed by PCR for the presence of X and Y chromosomes[65]. After separating the gonad and mesonephros, control (non-fluorescent) XY littermate gonads and XX or XY mesonephroi from *Nestin*-Cre; *Rosa*-Tomato embryos (detected via endogenous Tomato fluorescence under a fluorescent stereoscope) were recombined on 1.5% agar blocks[66]. The technique was modified such that the gonad–mesonephric border region was included in the mesonephros portion of the recombination, so that the nascent vasculature and Tomato-positive cells in the gonad–mesonephros border region were not disturbed during gonad–mesonephros separation[15]. All gonad–mesonephros recombinations were cultured for 48 h and processed for whole-mount immunofluorescence as described above.

**Cell counting**. To determine the percentage of Venus- or Tomato-positive LCs, we manually counted Venus/CYP17A1 or Tomato/CYP17A1 double-positive cells relative to CYP17A1 single-positive LCs, or Tomato/HSD17B3 double-positive cells relative to HSD17B3 single-positive LCs. For E13.5 experiments, whole-mount preparations were used, with four different confocal optical sections for each gonad, each separated by 10–12 μm to eliminate the possibility of double-counting cells. For E18.5 and adult experiments, 3–5 different cryosections, spaced apart by at least 60 μm to eliminate the possibility of double-counting cells, were used for each gonad. Venus or Tomato intensity was verified using threshold functions in NIS Elements software (Nikon). For all fetal experiments, $n = 3$ independent gonads from different embryos were used, while for ALC quantification, $n = 2$ independent males were used. Total cells counted were: $n = 264$ for E13.5 Venus, $n = 152$ for E13.5 Tomato, $n = 1574$ for E18.5 Venus, $n = 800$ for E18.5 Tomato, and $n = 644$ for adult Tomato. Graph results are shown as mean percentage ± SD.

**RNA extraction, cDNA synthesis, and quantitative PCR (qPCR)**. Total RNA was extracted from fetal testes cultured in the presence or absence of VEGFR-TKI II or γ-secretase inhibitor DAPT following 48 h in culture. The gonads were separated from the mesonephros in PBS and homogenized by vortexing in TRIzol reagent (Invitrogen/Thermo Fisher). A TRIzol/isopropanol precipitation method was used for total RNA extraction. cDNA synthesis was performed with 300–500 ng of total RNA, using an iScript cDNA synthesis kit (Bio-Rad). The cDNA was then subjected to qPCR using the Fast SYBR Green Master Mix (Applied Biosystems/Thermo Fisher) on the StepOnePlus Real-Time PCR System (Applied Biosystems/Thermo Fisher). The following parameters were used: 95 °C for 20 s, followed by 40 cycles of 95 °C for 3 s and 60 °C for 30 s, followed by a melt curve run. Primer specificity for a single amplicon was verified by melt curve analysis or agarose gel electrophoresis. *Gapdh* was used as an internal normalization control. Sequences of qPCR primers used are listed in Supplementary Table 2. Fold change in mRNA levels was calculated relative to controls using a ΔΔCt method. Results were shown as mean ± SD of $n = 3$ independent biological replicates (i.e., independent litters), each with ≥5 male gonads/litter. A two-tailed Student *t*-test was performed to calculate *P* values, in which $P < 0.05$ was considered statistically significant.

## Data availability

The data that support the findings of this study are available from the authors on reasonable request.

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

## Acknowledgements

We thank R. Kopan, J.-S. Park, R. Lang, and M. Nakafuku for mice; K. Kitamura and K. Morohashi for anti-ARX antibody; Y. Shima for anti-HSD17B3 antibody; R. Hegde for use of a hypoxic chamber; S. Potter for technical advice; and M. Kofron and M. Muntifering at the Cincinnati Children's Hospital Medical Center (CCHMC) Confocal Imaging Core. This work was supported by a CCHMC Research Innovation and Pilot Funding Grant; a CCHMC Trustee Grant Award; CCHMC developmental funds; a March of Dimes Basil O'Connor Starter Scholar Research Award (#5-FY14-32); and NIH grant #R35GM119458.

## Author contributions

D.K. conducted the majority of the experiments, performed data analyses, and co-wrote the manuscript. T.D. supervised the project, performed experiments, and co-wrote the manuscript.

## Additional information

**Competing interests:** The authors declare no competing interests.

