## [Peer Review File · Nature Communications]

Reviewers' comments:

Reviewer #1 (Remarks to the Author):

A review of the paper by Kumar and DeFalco entitled Perry vascular niche for multipotent progenitor's in the fetal testis

In this paper the authors describe the role that notch signalling plays in the development of fetal Leydig cells, showing that blood vessels are a critical component of the fetal Leydig cell niche. They also suggest that Leydig cell differentiation occurs by two mechanisms with different origins and requirements for notch signalling.

In general the experiments are completed to a high standard and the paper is well written. I have a few comments which focus on the mistaken paradigms that underpins the studies undertaken. I believe the results presented are an amalgamation of observations made in other studies. It is thus difficult to identify the novelty in this study. Specific comments below:

Can the authors comment on the observation that this current paper on the role of notch signalling in controlling the fetal versus adult Leydig cell population is similar to previous observations made in the paper of Liu et al, Development 143:3700-3710.

Line 14: whilst fetal Leydig cells do produce androgens, the accepted paradigm is that testosterone is produced through conversion of these precursor androgens in fetal Sertoli cells, see papers Shima et al; PMID: 23125070 and Baker PJ PMID: 9406858

line 24: Again in this case the authors seem to articulate an outdated paradigm as retention of fetal Leydig cells in the adult testis has been demonstrated. See paper Shima et al PMID: 26402718.

Line 145: 33% of cells that express tomato and derive from nestin expressing cells, would be consistent with nestin-expressing cells being the precursors of the adult Leydig cell population (or the retained FLC population described above). This would mean that rather than identifying a second population of fetal Leydig cells, the authors have identified an earlier origin of 30% of the adult Leydig cell population, which are known to be derived from Nestin precursors. Thus it is a semantic issue regarding what these cells are called, not a novel discovery. The key missing piece here is the failure to follow through the lineage tracing through to adulthood (due to this study being in culture). This would likely have confirmed consistency with the previously published literature.

line 28: The progenitor population that gives rise to fetal Leydig cells has been recently definitively identified. See paper Stevant et al; PMID 29425512

Reviewer #2 (Remarks to the Author):

The authors show that testis morphogenesis is influenced by a hypoxia-independent role of the specialized vasculature. The authors employ elegant lineage tracing with a developmental nestin-cre and a tamoxifen inducible nestin-cre to demonstrate that a subpopulation of various cell types seen at E18.5 are derived from the nestin-expressing population. They also use Notch reporter to determine which cell types were influenced by Notch signaling and they further show perturbations to the differentiation of fetal Leydig cells when either the vasculature was disrupted or Notch signaling was inhibited. The study is compelling in terms of understanding the origins of different fetal subpopulations and mechanistically on regarding their dependence on notch signaling. This

study is well executed given the technical complexities of this system. But enthusiasm is somewhat limited due to several technological limitations and also a question regarding the physiological significance of these cell populations.

Major points:

1. Temporal resolution of the tamoxifen inducible cre system - Using immunofluorescence, the authors demonstrate that nestin expression is turned on and off in different cell types between E13.5 and E18.5. It is difficult to utilize the nestin protein data to correlate with the tamoxifen inducible cre data because promoter-dependent cre induction could potentially occur prior to detectability of protein (ie, nestin "expression" could be more widespread at earlier time points than appreciated). Compared to developmental cre, use of Tamoxifen inducible cre in development is challenging for several reasons. There are a number of potentially confounding variables. First, the bioavailability of tamoxifen in a given organ will dictate the recombination efficiency and the duration of the time window in which recombination can occur, thus influencing the temporal resolution of the assay. An additional challenge regards the assumption that all cell types will reach steady state levels of reporter at the same time. This is complicated by the rate of cell division which is one of multiple variables that will determine when the reporter reaches a detectable level in a given cell type. While the authors briefly acknowledge the limitation regarding efficiency of recombination, there is no apparent effort dedicated to measure recombination efficiency. To this end, measurement of Nestin promoter activity using in situ hybridization could be useful.

2. Physiological significance - The authors characterize a subpopulation (1/3 of Leydig cells at E18.5 that are apparently derived from the nestin-expressing progenitors but there are no data to show the physiological significance of this population to subsequent outcomes related to development of the organ.

3. Specificity of VEGF receptor targeting - The use of VEGF receptor tyrosine kinase inhibitors might not be specific. These inhibitors might target non-VEGF receptor tyrosine kinases in non-vascular cells as well. Possibly the authors might consider employing neutralizing monoclonal antibodies to VEGF receptors which is more specific than small molecule tyrosine kinase inhibitors.

4. Specificity of Notch signaling - Could the authors define which Notch ligand, Dll4, Jagged1, Jagged2, or Dll1 etc.. might mediate Notch signaling? Similarly, it will be informative to identify the Notch receptor (Notch 2 or 3) that mediate the differentiation and maintenance of the Leydig cell progenitor cells.

Reviewer #3 (Remarks to the Author):

The origins of steroidogenic progenitors in the developing gonad and how they are maintained remain largely unknown. Kumar & DeFalco report in this manuscript a new role for vasculature in testicular differentiation and shed light on the origins of fetal Leydig cells (FLCs). Lineage tracing experiments and ex vivo analyses revealed that Nestin⁺ progenitors gave rise to some FLCs but also to peritubular myoid cells, vascular smooth muscle, and pericytes. They found that vasculature is important in maintaining the interstitial progenitor population that gives rise to differentiated Leydig cells. Active Notch signal in perivascular mesenchymal cells, driven by endothelial Notch ligands, maintains the Nestin⁺ progenitor pool. In the absence of vasculature, Nestin⁺ cells prematurely differentiate into Leydig cells.

This is a very interesting paper. Its strength arises from its detailed analysis of the fate of Nestin⁺

progenitors and ex-vivo experiments. It is well written and the figures are neat. The results support the conclusions and are of great interest for the scientific community working in the field of sex determination and testis development. Nevertheless, I feel that if a more thorough analysis is presented, this manuscript would be suitable for publication. It includes a better characterization of the Nestin-Cre tg lines, as well as the origin and the fate of the nestin+ progenitors in adult testis.

Major issues:

- 1) Fate of Nestin+ progenitors in post-pubertal testis. Several publications suggest that interstitial steroidogenic progenitors present in fetal testis retain their undifferentiated state during fetal stage and become adult Leydig cells in post-pubertal testis. The manuscript would not be complete with a cell lineage analysis in postnatal and adult testes. Since all transgenic lines have already been generated, it should not take too much time and energy to answer this question and it will increase the impact of the manuscript.
- 2) Origin of Nestin+ progenitors. Both supporting and interstitial cell lineages arise from WT1+ somatic progenitor pools in the gonadal primordium. What is the origin of these Nestin+ progenitors? Do they derive from the coelomic epithelium, do they express also Nr5a1/Ad4BP? In addition, a better characterization of these Nestin+ progenitor cells may be useful for example by analyzing the transcriptome of these cells isolated by FACS.
- 3) A better characterization of the Nestin-Cre and Nestin CreERT tg lines is required. In particular, the efficiency of recombination and penetrance of the two cre lines should be investigated. The number of Nestin positive cells in immunofluorescence in fig 4b (ctl E12.5) appears higher than the number of Tomato+ cells detected in fig 3a-b (ctl E13.5 and 18.5). This is important since the authors state that at E18.5, 33% of FLCs originate from Nestin-expressing progenitors. Is this a consequence of a low penetrance of the Cre? In addition, since tomato-positive cells were also present in the mesonephros, the author should evaluate/exclude the possibility that some of these mesonephros-derived cells Nestin+ cells are also FLCs progenitors. I suppose it may be possible for the authors to performed gonad WT/mesonephros Tomato-Nestin+ sandwiches and assess the fate of migrating tomato+ mesonephric cells.
- 4) I would like to have seen more insightful discussion and comparison with other recent studies. DeFalco et al, 2011, Myyabayashi et al 2013, Liu et al, 2016, Stevant et al 2018 have demonstrated the derivation of Sertoli cells and other cell types from a single WT1+ progenitor population (also SF1+) and the presence of multiple subpopulations of progenitors during sex determination. The studies by the Gao group further suggest that presence or absence of WT1 defines the divergence of Sertoli cell lineage and steroidogenic cell lineages (Wen et al 2014; Zhang et al 2015; Chen et al 2016). In the case of Liu et al 2016 and Stevant et al 2018, both mention a single origin for steroidogenic progenitors and that these progenitors restrict their fate during fetal development to acquire a steroidogenic progenitor identity. This information must be acknowledged and discussed in the introduction and/or in the discussion so that a fair assessment of the field could be provided to the readers.

Minor points:

- 1) Abstract, lines 10-13: "Additionally, our data strongly support a model in which Leydig cell differentiation occurs by at least two mechanisms, with each mechanism having unique progenitor origins and distinct requirements for Notch signaling to maintain the progenitor population." This is an overstatement since there is no clear evidence for the presence of two unique progenitor populations. This sentence should be modified.
- 2) Line 23: the sentence should be rephrased. "Distinct progenitor population" suggest that there are different progenitor populations for FLCs and ALCs.
- 3) Line 28: "The progenitor population (or populations) that gives rise to FLCs has not been definitively identified, but multiple putative progenitors for FLCs have been proposed 15". Several more recent publications should be also added as references such as Liu 2016, Stévant 2018, etc
- 4) Methods: authorization number(s) for animal experimentation should be included

5) Methods: a more complete description of the hypoxia experimental design and culture condition are needed. In particular how long did the organs stay in the hypoxic chamber? After how long were they harvested for qPCR?

Reviewer #1:

1. Can the authors comment on the observation that this current paper on the role of notch signalling in controlling the fetal versus adult Leydig cell population is similar to previous observations made in the paper of Liu et al, *Development* 143:3700-3710.

Our finding that disruption of Notch signaling leads to supernumerary fetal Leydig cells is consistent with the observations of Liu et al. 2016, *Development* 143:3700-3710 and Tang et al., 2008, *Development* 135(22):3745-3753. Along with Liu et al., 2016 we have also previously implicated Notch2 as the likely active Notch receptor in the testicular interstitial mesenchyme (DeFalco et al., 2013 *Biol Reprod* 88(4):91). However, none of these previous studies addressed the essential question of the ligand-presenting cell type required to activate Notch signaling in interstitial cells to maintain their progenitor status. Therefore, our finding that vasculature is the critical cell type for activating Notch signaling to maintain Leydig progenitors is a significant and novel finding in the field. We would also like to point out that CreER lines used for lineage tracing in Liu et al., 2016, *Hes1*-CreER and *Gli1*-CreER, are very broadly expressed in the interstitium of the fetal testis, which precluded any analyses of specific interstitial cell populations, such as perivascular cells. Our use of *Nestin*-CreER allows for a specific and definitive targeting of perivascular cells to determine their fate within the testis and the role of Notch signaling within these cells, making this study unique as compared to previous publications.

As for the question of the regulation of fetal versus adult Leydig cells, we did not address this point in the original version of this manuscript; however, we have addressed this point in the revision and demonstrated, via lineage tracing into adulthood, that *Nestin*-expressing cells not only give rise to fetal Leydig cells, but also adult Leydig cells. Using an anti-HSD17B3 antibody, which specifically labels adult Leydig cells, as opposed to retained fetal Leydig cells which do not express this marker, we found that *Nestin*-expressing cells give rise to a subset of both fetal and adult Leydig cells. We have added this new data as a supplementary figure (Supplementary Fig. 6).

As for the role of Notch signaling in regulating adult Leydig cells, in a previous publication we used qPCR and transgenic Notch reporter mice to show that Notch signaling is likely not active in the adult testis interstitium (DeFalco et al., 2013 *Biol Reprod* 88(4):91). Therefore, we think that Notch signaling is actively regulating only fetal Leydig cell differentiation. However, it is likely that Notch signaling is required to maintain fetal interstitial cells in an undifferentiated state, so that they can differentiate into adult Leydig cells later on in development, as was proposed by Liu et al. 2016.

2. Line 14: whilst fetal Leydig cells do produce androgens, the accepted paradigm is that testosterone is produced through conversion of these precursor androgens in fetal Sertoli cells, see papers Shima et al; PMID: 23125070 and Baker PJ PMID: 9406858

We thank the reviewer for pointing out this omission. We have clarified this point in the Introduction on lines 29-33 and have added the suggested references.

3. line 24: Again in this case the authors seem to articulate an outdated paradigm as retention of fetal Leydig cells in the adult testis has been demonstrated. See paper Shima et al PMID: 26402718.

We agree that these findings should be mentioned. We have modified the discussion of this topic by including this new information, along with 2 references, in the Introduction on lines 23-28.

4. Line 145: 33% of cells that express tomato and derive from nestin expressing cells, would be consistent with nestin-expressing cells being the precursors of the adult Leydig cell population (or the retained FLC population described above). This would mean that rather than identifying a second population of fetal Leydig cells, the authors have identified an earlier origin of 30% of the adult Leydig cell population, which are known to be derived from Nestin precursors. Thus it is a semantic issue regarding what these cells are called, not a novel discovery. The key missing piece here is the failure to follow through the lineage tracing through to adulthood (due to this study being in culture). This would likely have confirmed consistency with the previously published literature.

We agree with the reviewer that this is an important experiment. We performed Tomato lineage tracing through adulthood to address the fate of *Nestin*-expressing cells in the adult testis. Using a HSD17B3 antibody to specifically label adult Leydig cells (as opposed to retained fetal Leydig cells in the adult testis, which do not express HSD17B3), we observed that *Nestin*-expressing cells give rise to a subset of HSD17B3-positive adult Leydig cells, as well as a subset of retained fetal Leydig cells in the adult testis. We have included this data as a new supplementary figure (Supplementary Fig. 6).

5. line 28: The progenitor population that gives rise to fetal Leydig cells has been recently definitively identified. See paper Stevant et al; PMID 29425512

We thank the reviewer for pointing out this omission. We have included a mention of this reference in the Introduction on lines 38-39 and Discussion sections on lines 390-394. However, as a caveat, this publication characterized the transcriptome of a general early progenitor population and an interstitial progenitor population, but did not specifically address specific progenitors for fetal Leydig cells (or any other specific interstitial cell types). Also, this study only analyzed cells expressing *Nr5a1*-GFP, which may not include all interstitial/mesenchymal cells in the gonad; in fact, only a few differentiated Leydig cells were obtained using this method. Also, our newly added data in Supplementary Fig. 4 shows that *Nestin*-expressing progenitor cells do not express NR5A1, so this population was likely not included in the analyses of Stevant et al., 2018.

Reviewer #2:

1. Temporal resolution of the tamoxifen inducible cre system - Using immunofluorescence, the authors demonstrate that nestin expression is turned on and off in different cell types between E13.5 and E18.5. It

is difficult to utilize the nestin protein data to correlate with the tamoxifen inducible cre data because promoter-dependent cre induction could potentially occur prior to detectability of protein (ie, nestin “expression” could be more widespread at earlier time points than appreciated). Compared to developmental cre, use of Tamoxifen inducible cre in development is challenging for several reasons. There are a number of potentially confounding variables. First, the bioavailability of tamoxifen in a given organ will dictate the recombination efficiency and the duration of the time window in which recombination can occur, thus influencing the temporal resolution of the assay. An additional challenge regards the assumption that all cell types will reach steady state levels of reporter at the same time. This is complicated by the rate of cell division which is one of multiple variables that will determine when the reporter reaches a detectable level in a given cell type. While the authors briefly acknowledge the limitation regarding efficiency of recombination, there is no apparent effort dedicated to measure recombination efficiency. To this end, measurement of Nestin promoter activity using *in situ* hybridization could be useful.

These are intriguing questions brought up by the reviewer. To address the issue of potential dynamic expression of *Nestin* promoter activity, we used a tamoxifen-inducible system to specifically target *Nestin*-expressing cells at a time when Nestin is only expressed in perivascular cells of the fetal testis. We compared Cre or CreER activity to Nestin protein for purposes of assessing specificity, rather than efficiency, since for lineage tracing purposes, it is often not necessary to obtain 100% labeling efficiency. However, our images in Figs. 2a, 2g, and 3c clearly show that *Nestin*-Cre and *Nestin*-CreER specifically target perivascular cells and effectively recapitulate Nestin’s expression pattern at E13.5 with high efficiency and specificity. We also examined E18.5 testes from *Nestin*-Cre;*Rosa*-Tomato embryos, and found that >90% of Nestin-immunopositive cells were labeled with Tomato, suggesting that *Nestin*-Cre is efficient at targeting Nestin-expressing cells. Regardless of Cre efficiency, we were more interested in effectively labeling and lineage-tracing perivascular cells than reproducing the *Nestin* expression pattern with absolute timing and precision. We concede, as the reviewer has mentioned, that there are some limitations to the tamoxifen-inducible CreER system. However, due to: the short half-life of 4-hydroxytamoxifen (6 hours in mice; Robinson et al. Drug Metab Dispos. 1991;19(1):36-43); the small size and extensive vascularization of the fetal testis; rapid folding rate (~1h; Shaner et al., Nat Biotechnol. 2004;22(12):1567-72) and high quantum yield of fluorescence for tdTomato; and highly sensitive confocal imaging technologies, we are confident that detectable levels of reporter expression occur quickly (less than 24 hours) and uniformly in the gonad. We are unsure as to how *in situ* hybridization for *Nestin* activity will be informative because, as the reviewer has pointed out, there is likely a lag between *Nestin* promoter activity and recombination leading to detectable reporter expression, and the lag will likely be even greater at the RNA level as compared to the protein level.

2. *Physiological significance* - The authors characterize a subpopulation (1/3 of Leydig cells at E18.5 that are apparently derived from the nestin-expressing progenitors but there are no data to show the physiological significance of this population to subsequent outcomes related to development of the organ.

We agree that some physiological context with respect to this cell population would be informative. To address this point, we have performed lineage tracing to adulthood and assessed the specific fate of Nestin-expressing cells. We have determined that these cells give rise to a subset of adult Leydig cells and retained fetal Leydig cells in the adult testis, as well as other cell types (also please see our responses to comments #1 and #4 for Reviewer 1 above). We have included this data in a new Supplementary Fig. 6.

3. *Specificity of VEGF receptor targeting* - The use of VEGF receptor tyrosine kinase inhibitors might not be specific. These inhibitors might target non-VEGF receptor tyrosine kinases in non-vascular cells as

well. Possibly the authors might consider employing neutralizing monoclonal antibodies to VEGF receptors which is more specific than small molecule tyrosine kinase inhibitors.

We agree that small-molecule tyrosine kinase inhibitors could potentially have unintended, off-target effects. To address this concern, we have included new data using a peptide-based Aflibercept/VEGF-Trap, which is extremely specific to VEGFA (with picomolar affinity), and is specifically delivered to the vasculature via fetal heart injection. Aflibercept-treated testes exhibited a very similar phenotype to VEGFR-TKI-II-treated testes, in which Aflibercept-treated testes contained supernumerary Leydig cells. We have added this new data as Supplementary Fig. 8. Additionally, we have used other small-molecule inhibitors, such as axitinib, which are most effective against VEGF receptors, and seen very similar results via qPCR and immunofluorescence (not shown). Finally, our qPCR data in Supplementary Fig. 1 suggests that Sertoli cells are not affected by VEGFR-TKI II treatment, and is consistent with its effects being specific to the vasculature. In all, these findings strongly suggest that the observed testicular phenotypes are specifically due to inhibition of VEGF-induced vascularization, as opposed to off-target effects or effects on other cell types.

4. Specificity of Notch signaling - Could the authors define which Notch ligand, Dll4, Jagged1, Jagged2, or Dll1 etc.. might mediate Notch signaling? Similarly, it will be informative to identify the Notch receptor (Notch 2 or 3) that mediate the differentiation and maintenance of the Leydig cell progenitor cells.

We agree that identifying the specific Notch ligand(s) and Notch receptor(s) that mediate Notch signaling would be informative. To address this point, we have performed immunofluorescence and Notch-activity lineage-tracing experiments. As for the ligand, we have focused on *Dll4*, which according to our previous transcriptome analyses (Jameson et al., 2012 *PLoS Genetics*) is expressed specifically and at the highest level in endothelial cells. We have included new DLL4 immunofluorescence data in Supplementary Fig. 9, which reveals DLL4 expression specifically in fetal testicular endothelial cells adjacent to Nestin-positive cells. We have also included new Notch-activity lineage tracing data in Supplementary Fig. 9 using a *Notch2-ICD-Cre* line (Liu et al., 2013 *Dev Cell*. 25(6):585-98), in which cells that have undergone active NOTCH2 signaling (and their progeny) are permanently labeled with YFP; these experiments revealed that perivascular cells undergo active Notch signaling through NOTCH2, which is consistent with previous conditional loss-of-function experiments showing a requirement for *Notch2* in maintaining Leydig progenitors (Liu et al. 2016. *Development* 143:3700-3710).

Reviewer #3:

1. Fate of Nestin+ progenitors in post-pubertal testis. Several publications suggest that interstitial steroidogenic progenitors present in fetal testis retain their undifferentiated state during fetal stage and become adult Leydig cells in post-pubertal testis. The manuscript would not be complete with a cell lineage analysis in postnatal and adult testes. Since all transgenic lines have already been generated, it should not take too much time and energy to answer this question and it will increase the impact of the manuscript.

This point has been brought up by the other reviewers (please see comments 1 and 4 by Reviewer #1 and comment 2 by Reviewer #2), and we concur that this analysis is necessary. To address this point, we have included a new Supplementary Fig. 6 in which we show lineage tracing data into adulthood and

have found that *Nestin*-positive cells give rise to both fetal and adult Leydig cells, as well as peritubular myoid cells and undifferentiated perivascular cells, such as pericytes, in the adult testis.

2. Origin of Nestin+ progenitors. Both supporting and interstitial cell lineages arise from WT1+ somatic progenitor pools in the gonadal primordium. What is the origin of these Nestin+ progenitors? Do they derive from the coelomic epithelium, do they express also Nr5a1/Ad4BP? In addition, a better characterization of these Nestin+ progenitor cells may be useful for example by analyzing the transcriptome of these cells isolated by FACS.

We agree that more analysis of the origin of *Nestin*-positive cells would be informative. To address whether *Nestin*-lineage-labeled cells arise from the mesonephros, we performed a gonad-mesonephros recombination culture assay with Tomato-labeled *Nestin*-expressing cells in the mesonephros. Additionally, we performed a dye-labeling experiment showing that *Nestin*-expressing cells are not derived from the coelomic epithelium, which is consistent with a mesonephric origin for *Nestin*-positive cells. These data are now included in a new Fig. 3. We have also included new results in Supplementary Fig. 4 in which we co-stained *Nestin*-positive cells with NR5A1 (SF1) and found that undifferentiated *Nestin*-positive cells do not express NR5A1, unlike virtually all other somatic non-endothelial cells in the gonad. Therefore, our data suggests these cells are a unique gonadal cell population that is likely derived from migrating mesonephric cells. However, we also found that *Nestin*-positive cells express WT1 (see Supplementary Fig. 4), and are likely a subset of the parental WT1-positive gonadal somatic progenitor population at E10.5 that was described by Liu et al. 2016, *Development* 143:3700-3710.

3. A better characterization of the Nestin-Cre and Nestin CreERT tg lines is required. In particular, the efficiency of recombination and penetrance of the two cre lines should be investigated. The number of Nestin positive cells in immunofluorescence in fig 4b (ctl E12.5) appears higher than the number of Tomato+ cells detected in fig 3a-b (ctl E13.5 and 18.5). This is important since the authors state that at E18.5, 33% of FLCs originate from Nestin-expressing progenitors. Is this a consequence of a low penetrance of the Cre? In addition, since tomato-positive cells were also present in the mesonephros, the author should evaluate/exclude the possibility that some of these mesonephros-derived cells Nestin+ cells are also FLCs progenitors. I suppose it may be possible for the authors to performed gonad WT/mesonephros Tomato-Nestin+ sandwiches and assess the fate of migrating tomato+ mesonephric cells.

These are important questions brought up by the reviewer. The discrepancy between the number of *Nestin*-positive cells in the previous Fig. 4b and Tomato-positive cells in the previous Fig. 3a is likely due to the fact that the gonads in Fig. 4 were cultured, whereas the gonads in Fig. 3 developed *in utero*; we have observed that vasculature, and thus *Nestin*+ cells, develop slightly differently in cultured gonads. If one compares the number of *Nestin*-positive cells in Fig. 2a versus the number of Tomato+ cells in Fig. 2g/3a (*Nestin*-Cre) and 3c (*Nestin*-CreER), which are all from gonads developed *in utero*, the number of cells is similar. Our examination of *Nestin*-Cre activity compared to *Nestin* protein expression indicates that *Nestin*-Cre is at least, if not more than, 90% efficient (please see comment 1 from Reviewer #2 above). Therefore, the 33% number referred to by the reviewer is likely not due to a low efficiency of Cre. Instead, it is likely due to the fact that many Leydig cells, as we have previously shown (DeFalco et al., 2011 *Dev Biol* 352(1):14-26), are derived from the coelomic epithelium, and not from *Nestin*+ cells, which our newly added data shows are not derived from the coelomic epithelium.

To address the second question of whether some of these *Nestin*+ cells are mesonephros-derived, we have performed wild-type gonad/Tomato-*Nestin*+ mesonephros recombination culture assays and

observed Tomato+ cells migrating throughout the gonad, including reaching the vicinity of the coelomic artery; this data is included in a new Fig. 3. These data suggest that at least a subset of Nestin+-positive cells migrate from the mesonephros into the gonad. Given that we have tamoxifen-labeled *Nestin-CreER;Rosa-Tomato* gonads at E11.5, when virtually all Tomato+ cells are in the mesonephros (Fig. 2f), it is possible that most, if not all, Nestin+ cells in the gonad originally migrated from the mesonephros at the time of initial testicular vascular remodeling. However, we could not assess directly whether these particular mesonephric-derived migrating Nestin+ cells give rise to FLCs in our recombination culture assays, since early (E13.5) Tomato-positive cells do not yet give rise to FLCs (see Fig. 4e), and the culture experiments are only viable for about 48 hours after starting them at E11.5-E12.0.

4. I would like to have seen more insightful discussion and comparison with other recent studies. DeFalco et al, 2011, Myyabayashi et al 2013, Liu et al, 2016, Stevant et al 2018 have demonstrated the derivation of Sertoli cells and other cell types from a single WT1+ progenitor population (also SF1+) and the presence of multiple subpopulations of progenitors during sex determination. The studies by the Gao group further suggest that presence or absence of WT1 defines the divergence of Sertoli cell lineage and steroidogenic cell lineages (Wen et al 2014; Zhang et al 2015; Chen et al 2016). In the case of Liu et al 2016 and Stevant et al 2018, both mention a single origin for steroidogenic progenitors and that these progenitors restrict their fate during fetal development to acquire a steroidogenic progenitor identity. This information must be acknowledged and discussed in the introduction and/or in the discussion so that a fair assessment of the field could be provided to the readers.

We concur with the reviewer that a discussion of these references is needed to provide a context for our results. As requested, we have added a significant discussion of the aforementioned relevant publications to the Introduction on lines 36-42 and the Discussion on lines 378-394.

Minor points:

1. Abstract, lines 10-13: "Additionally, our data strongly support a model in which Leydig cell differentiation occurs by at least two mechanisms, with each mechanism having unique progenitor origins and distinct requirements for Notch signaling to maintain the progenitor population." This is an overstatement since there is no clear evidence for the presence of two unique progenitor populations. This sentence should be modified.

We feel that our newly added data, including tissue recombination assays, coelomic epithelial labeling, and NR5A1/SF1 immunofluorescence, is consistent with Nestin-positive cells having a unique mesonephric origin as compared to other gonadal cells that are derived from the coelomic epithelium; therefore, we believe that we now have more evidence supporting the idea of 2 unique progenitor populations. However, to soften the language, we have replaced the words "by at least two mechanisms" with "by at least two means", which is less assertive in terms of differences in molecular regulation of these 2 methods of fetal Leydig cell development.

2. Line 23: the sentence should be rephrased. "Distinct progenitor population" suggest that there are different progenitor populations for FLCs and ALCs.

We agree that this phrase could be misleading. We have explained this topic in more detail and have modified this section of the Introduction to include other literature in the field (lines 23-42).

3. Line 28: "The progenitor population (or populations) that gives rise to FLCs has not been definitively identified, but multiple putative progenitors for FLCs have been proposed 15". Several more recent publications should be also added as references such as Liu 2016, Stévant 2018, etc.

We thank the reviewer for pointing out these omissions. We have modified the text and have now included these references in the Introduction sections on lines 38-42 and in the Discussion section on lines 381-394.

4. Methods: authorization number(s) for animal experimentation should be included.

As requested, we have now included the authorization numbers for our IACUC (institutional animal care and use committee) experimentation protocols in the Methods section (lines 488-489).

5. Methods: a more complete description of the hypoxia experimental design and culture condition are needed. I particular how long did the organs stay in the hypoxic chamber? After how long were they harvested for qPCR?

As requested, we have now included this additional information as a new section within the Methods (lines 536-541). We have also clarified in the legend for Supplementary Fig. 2a that the qPCR analyses in that figure were performed on gonads cultured under normoxic conditions for 48 hours, not under hypoxic conditions. We apologize for any confusion.

REVIEWERS' COMMENTS:

Reviewer #1 (Remarks to the Author):

my questions largely related to the fate of the developing cells. The lineage tracing into adulthood has been an important addition to the paper which I believe has improved the conclusions. My other issues related to the context in which the study was framed; again I think this as been improved through the addition of relevant citations.

Reviewer #2 (Remarks to the Author):

The authors have adequately responded to the majority of my concerns.

The only remaining minor concerns are that while the authors did add the adult lineage tracing data, there are no functional experiments to determine the importance of this cell subpopulation to homeostatic or stressed adult organ function, which is still a limitation. Another limitation is that the authors do not quantify the numbers of the different adult cell types derived from the nestin+ precursors.

Reviewer #3 (Remarks to the Author):

The authors have made a thorough and sincere effort to address the reviewers' comments. The addition of new data and text have improved the manuscript significantly. There are no outstanding scientific concerns.

There is also a typo on line 24; a parenthesis is missing.

Reviewer #1:

1. My questions largely related to the fate of the developing cells. The lineage tracing into adulthood has been an important addition to the paper which I believe has improved the conclusions. My other issues related to the context in which the study was framed; again I think this has been improved through the addition of relevant citations.

We appreciate the reviewer's comments, which we found helpful in our efforts to improve the manuscript.

Reviewer #2:

1. The authors have adequately responded to the majority of my concerns.

We would like to thank the reviewer for his/her comments.

The only remaining minor concerns are that while the authors did add the adult lineage tracing data, there are no functional experiments to determine the importance of this cell subpopulation to homeostatic or stressed adult organ function...

We agree with the reviewer's minor concern that this question may be interesting, and we will certainly consider it for our future studies. However, we feel that the role of these cells in adult organ function is beyond the scope of the current study, which focuses on a perivascular niche for fetal Leydig progenitor cells.

2. ...the authors do not quantify the numbers of the different adult cell types derived from the nestin+ precursors.

We agree with the reviewer that some quantification of the adult cell types derived from Nestin-expressing precursors would be informative. We performed cell counts of adult Leydig cells within *Nestin-Cre; Rosa-Tomato* adult testes using an anti-HSD17B3 antibody, and found that 47.2% of adult Leydig cells were Tomato-positive, i.e., derived from Nestin+ precursors. We have included this new data in the Results section in the last paragraph of the section entitled "Nestin-positive cells give rise to a subset of FLCs and smooth muscle."

Reviewer #3:

1. The authors have made a thorough and sincere effort to address the reviewers' comments. The addition of new data and text have improved the manuscript significantly. There are no outstanding scientific concerns.

There is also a typo on line 24; a parenthesis is missing.

We thank the reviewer for his/her helpful comments. We have corrected the error.